# Distributional Preference Alignment of LLMs via Optimal Transport

Igor Melnyk*, Youssef Mroueh*, Brian Belgodere*, Mattia Rigotti, Apoorva Nitsure,

Mikhail Yurochkin, Kristjan Greenewald, Jiri Navratil, and Jarret Ross

IBM Research
MIT-IBM Watson AI Lab

## Abstract

Current LLM alignment techniques use pairwise human preferences at a sample level, and as such, they do not imply an alignment on the distributional level. We propose in this paper Alignment via Optimal Transport (AOT), a novel method for distributional preference alignment of LLMs. AOT aligns LLMs on unpaired preference data by making the reward distribution of the positive samples stochastically dominant in the first order on the distribution of negative samples. We introduce a convex relaxation of this first-order stochastic dominance and cast it as an optimal transport problem with a smooth and convex cost. Thanks to the one-dimensional nature of the resulting optimal transport problem and the convexity of the cost, it has a closed-form solution via sorting on empirical measures. We fine-tune LLMs with this AOT objective, which enables alignment by penalizing the violation of the stochastic dominance of the reward distribution of the positive samples on the reward distribution of the negative samples. We analyze the sample complexity of AOT by considering the dual of the OT problem and show that it converges at the parametric rate. Empirically, we show on a diverse set of alignment datasets and LLMs that AOT leads to state-of-the-art models in the 7B family of models when evaluated with Open LLM Benchmarks and AlpacaEval. Code for AOT is available in the Hugging Face TRL library `https://ibm.biz/AOT_TRL`.

## 1 Introduction

Aligning Large Language Models (LLMs) with human preferences is a crucial step in making these models safe and having them follow instructions faithfully. By ensuring that LLMs adhere to human preferences, values, ethics, and desired behaviors we can reduce the risk of generating harmful, biased, or inappropriate content.

Reinforcement Learning from Human Feedback, *RLHF* [Christiano et al., 2017, Stiennon et al., 2020, Ouyang et al., 2022, Bai et al., 2022], achieves this by learning a reward model on human preference data, followed by fine-tuning the LLM to maximize the reward score while staying close to the initial reference policy to retain utility from the pre-trained model. Recently, new paradigms departed from RLHF towards direct preference optimization methods such as DPO [Rafailov et al., 2024], SLIC [Zhao et al., 2023], and Identity Policy optimization [Azar et al., 2024]. In these approaches, the reward is expressed in terms of the log-likelihood ratio between the LLM policy and the reference model. The training is done on paired preference data, i.e. as triplets of prompts, chosen and rejected sentences, where for each prompt a chosen and a rejected sample are available. The training objective is to maximize the margin between the log-likelihood ratio evaluated on the chosen sentence versus

---

*Now with Capital One; Work done while at IBM Research

38th Conference on Neural Information Processing Systems (NeurIPS 2024).

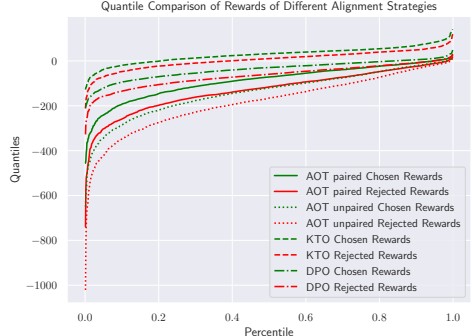 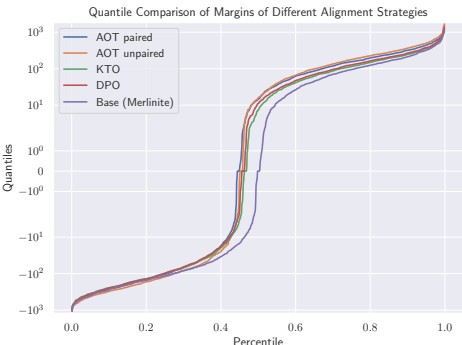

(a) Stochastic Dominance of Reward of Chosen on Rejected: AOT achieves a larger margin between the quantile plots of chosen and rejected rewards.

(b) Stochastic Dominance of AOT's optimized policy margin (between Chosen on Rejected) on the margin of the reference policy.

Figure 1: AOT in the paired & unpaired settings enables first-order stochastic dominance of the chosen reward distribution on the rejected distribution (a). The margin between the quantiles of chosen and rejected rewards is larger than alternative strategies. In (b), we see that AOT's policy chosen to rejected log-likelihood ratio dominates that ratio for the base model and alternative strategies.

the log-likelihood ratio on rejected sentences. When paired preference data is not available, and the preference data instead takes the form of distinct marginals of chosen prompt/response pairs and rejected prompt/response pairs, we refer to this setup as the unpaired data setting. Ethayarajh et al. [2024] used Kahneman & Tversky's prospect theory in the unpaired setting and proposed the KTO method that maximizes the margin between the chosen reward and the average reward of rejected sentences and pushes the reward of a rejected sentence below the average reward of chosen sentences.

In this paper, we introduce a new *distributional* optimization method for fine-tuning LLMs from human preference data. Previous work in the paired setting focused on improving the reward of chosen sentences over rejected sentences on a per-sample basis. This procedure does not lead to a preference on a distributional level of the chosen marginal on the rejected marginal. In probabilistic terms, we would like to induce stochastic dominance of the reward of chosen sentences on the reward of rejected ones. First order Stochastic Dominance (FSD, see e.g. Ogryczak and Ruszczynski, 2002) of a random variable $X$ on a random variable $Y$, means that all quantiles values of $X$ are larger than those of $Y$. Our main contribution is introducing AOT, *Alignment via Optimal Transport*, a new method that enables distributional alignment. We do so by devising a new AOT objective function that induces in the unpaired setting FSD dominance of *chosen reward's* distribution over *rejected reward's* distribution. We call this unpaired variant uAOT. In the paired setting, we introduce pAOT that encourages a dominance of chosen to rejected log likelihood ratio of the optimized policy on that ratio for the reference base policy. We show that the AOT cost can be cast as a one-dimensional optimal transport problem that can be solved via sorting and efficiently optimized for the LLM. AOT enjoys also nice statistical proprieties and achieves the parametric rate since its objective can be seen as a smooth one-dimensional optimal transport problem. AOT achieves state-of-the-art results on the Alpaca leaderboard [Dubois et al., 2024] using the `Merlinite 7B` model [Sudalairaj et al., 2024] as a base and scores as the highest 7B model at the time of writing this paper.

To introduce the important concepts of our work pictorially, we show in Fig. 1a the quantile plots of the rewards of AOT and alternative alignment strategies (DPO, KTO) for chosen responses (in green) and rejected responses in (red). The quantile plots are estimated on a paired test set. We see that AOT leads to chosen rewards that have larger margins than those of rejected rewards across all percentiles. More importantly, this margin is larger in AOT models than in policies coming from alternative alignment strategies. We then show in Fig. 1b how the AOT aligned policy's chosen-to-rejected log-likelihood ratio dominates that same ratio evaluated on the base model's ratio across all percentiles. The distributional alignment induced by AOT ensures a large margin between all quantiles so that the preference is reflected not only on average but distributionally. We formalize distributional preference in the next section.

## 2 Distributional Preference via First Order Stochastic Dominance

**First Order Stochastic Dominance** For a real random variable $Z$ we denote $F_Z^{(-1)} : [0,1] \to \overline{\mathbb{R}}$ the left-continuous inverse of the Cumulative Distribution Function (CDF) $F_Z$:

$$Q_Z(p) = F_Z^{(-1)}(p) = \inf\{\eta : F_Z(\eta) \geq p\} \text{ for } p \in [0,1].$$

Given two random variables $Z_1$ and $Z_2$, we say that $Z_1$ dominates $Z_2$ in the first order if $Z_1$ has larger quantiles than $Z_2$ for all percentiles $p$:

$$Z_1 \underset{\text{FSD}}{\succcurlyeq} Z_2 \iff Q_{Z_1}(p) \geq Q_{Z_2}(p), \quad \forall p \in [0,1]. \tag{1}$$

Let $\mathcal{X}$ be the space of prompts $X$ and $\mathcal{Y}$ be the space of responses $Y$ from an LLM conditioned on a prompt $X \in \mathcal{X}$. The reference LLM is represented as policy $\pi_{\text{ref}}(Y|X)$, i.e., as a conditional probability on $\mathcal{Y}$ given a prompt $X \in \mathcal{X}$. We note the LLM policy we are optimizing by $\pi_\theta$ where $\theta$ is a parameter belonging to a bounded parameter space $\Theta \subset \mathbb{R}^{d_\theta}$. For a measure $\mu \in \mathcal{P}(\mathcal{X} \times \mathcal{Y})$ and a mapping $r : \mathcal{X} \times \mathcal{Y} \to \mathbb{R}$, we note as $r_\sharp \mu$ the pushforward map of $\mu$ through $r$. In particular, for empirical measures $\mu = \frac{1}{n} \sum_{i=1}^n \delta_{(x_i, y_i)}$, we have that $r_\sharp \mu = \frac{1}{n} \sum_{i=1}^n \delta_{r(x_i, y_i)}$.

**DPO as a Pointwise Preference Approach** In Direct Preference Optimization (DPO, Rafailov et al., 2024), the reward being optimized by the LLM has the following form :

$$r_\theta(x, y) = \beta \log \frac{\pi_\theta(y|x)}{\pi_{\text{ref}}(y|x)} + \beta \log(Z(x)),$$

where $Z(x)$ is a normalization constant. DPO assumes access to a paired preference dataset $(X, Y_+, Y_-) \sim \mu$ where $Y_+$ denotes a positive (chosen) response to which we would like to assign a high reward, and $Y_-$ a negative (rejected) response to which we would like to assign a low reward. This can be formalized as minimizing the logarithmic sigmoid loss :

$$\min_{\theta \in \Theta} -\mathbb{E}_{(x, y_+, y_-) \sim \mu} \log(\sigma(\beta(r_\theta(x, y_+) - r_\theta(x, y_-)))),$$

and since the difference is taken for the same $x$, the normalization $Z(x)$ disappears resulting in:

$$\min_\theta -\mathbb{E}_{(x, y_+, y_-) \sim \mu} \log\left(\sigma\left(\beta \log\left(\frac{\pi_\theta(y_+|x)}{\pi_{\text{ref}}(y_+|x)}\right) - \beta \log\left(\frac{\pi_\theta(y_-|x)}{\pi_{\text{ref}}(y_-|x)}\right)\right)\right).$$

We can interpret this as a pointwise constraint inducing preference for positive over negative reward outcomes as follows:

$$\log\left(\frac{\pi_\theta(y_+|x)}{\pi_{\text{ref}}(y_+|x)}\right) \geq \log\left(\frac{\pi_\theta(y_-|x)}{\pi_{\text{ref}}(y_-|x)}\right), \quad \forall (x, y_+, y_-) \sim \mu. \tag{2}$$

DPO can then be interpreted as a relaxation of this constraint through the logistic loss, which also suggests other preference optimization algorithms through relaxations using, for example, the hinge loss as proposed in SLIC [Zhao et al., 2023].

### 2.1 Distributional Preference via Stochastic Dominance

Our main insight from looking at the pointwise constraint in Eq. (2) is that we can recast it as a distributional constraint in terms of stochastic dominance of the random variable $Z_\theta^+ = \log(\frac{\pi_\theta(Y_+|X)}{\pi_{\text{ref}}(Y_+|X)})$ of positive outcomes on the random variable $Z_\theta^- = \log(\frac{\pi_\theta(Y_-|X)}{\pi_{\text{ref}}(Y_-|X)})$ of negative outcomes. This is especially valuable in the unpaired setting without access to triplets of prompts and positive and negative responses as required by DPO. This is indeed the same setting considered by KTO [Ethayarajh et al., 2024]. The following paragraph formalizes this unpaired distributional preference.

**Distributional Unpaired Preference** We assume here that we don't have access to triplets of prompts and positive/negative responses $(x, y_+, y_-)$. Instead, we assume separate access to $\mu_+ \in \mathcal{P}(\mathcal{X} \times \mathcal{Y})$, a distribution of positive prompt/response pairs $(X_+, Y_+)$ we would like to be highly rewarded and reinforce in the policy, and $\mu_- \in \mathcal{P}(\mathcal{X} \times \mathcal{Y})$ the distribution of the negative samples $(X_-, Y_-)$ to be associated with low reward. We define the distributional preference as follows:

**Definition 1** (Distributional Preference in the Unpaired Setting). *A policy $\pi$ prefers distributionally $\mu_+$ on $\mu_-$ with respect to a reference policy $\pi_{\text{ref}}$ if:*

$$\log \frac{\pi_\theta(Y_+|X_+)}{\pi_{\text{ref}}(Y_+|X_+)} \underset{FSD}{\succcurlyeq} \log \frac{\pi_\theta(Y_-|X_-)}{\pi_{\text{ref}}(Y_-|X_-)}.$$

*In other words, noting $r_u \circ \pi_\theta(x,y) = \log \frac{\pi_\theta(y|x)}{\pi_{\text{ref}}(y|x)}$, the distributional preference in the unpaired setting means that we have the following constraint:*

$$(r_u \circ \pi_\theta)_\sharp \mu_+ \underset{FSD}{\succcurlyeq} (r_u \circ \pi_\theta)_\sharp \mu_-. \tag{3}$$

**Distributional Paired Preference**  Note that we can rewrite Eq. (2) in the equivalent form:

$$\log \frac{\pi_\theta(y_+|x)}{\pi_\theta(y_-|x)} \geq \log \frac{\pi_{\text{ref}}(y_+|x)}{\pi_{\text{ref}}(y_-|x)}, \quad \forall (x, y_+, y_-) \sim \mu. \tag{4}$$

In order to turn this into a distributional constraint we need access to a paired preference dataset as in DPO $(X, Y_+, Y_-) \sim \mu$, and impose stochastic dominance of the random variable $Z_\theta = \log \frac{\pi_\theta(Y_+|X)}{\pi_\theta(Y_-|X)}$ indexed by the policy we are optimizing on the random variable $Z_{\text{ref}} = \log \frac{\pi_{\text{ref}}(Y_+|X)}{\pi_{\text{ref}}(Y_-|X)}$ indexed by the reference policy. $Z_\theta$ and $Z_{\text{ref}}$ represent here the $\log$ likelihood ratio of positive to negative outcome under the policies $\pi_\theta$ and $\pi_{\text{ref}}$, respectively. Hence, it is desirable to constrain the policy $\pi_\theta$ to have a larger excess log probability between positive and negative outcomes than that resulting from the reference policy $\pi_{\text{ref}}$.

We define below more formally the paired distributional preference via stochastic dominance:

**Definition 2** (Distributional Preference in the Paired Setting). *We say that the policy $\pi_\theta$ distributionally dominates $\pi_{\text{ref}}$ in terms of $\log$ probability ratio of positive and negative responses if:*

$$\log \frac{\pi_\theta(Y_+|X)}{\pi_\theta(Y_-|X)} \underset{FSD}{\succcurlyeq} \log \frac{\pi_{\text{ref}}(Y_+|X)}{\pi_{\text{ref}}(Y_-|X)}.$$

*Noting $r_p \circ \pi_\theta(x, y_+, y_-) = \log \frac{\pi_\theta(y_+|x)}{\pi_\theta(y_-|x)}$ this can be written as follows:*

$$(r_p \circ \pi_\theta)_\sharp \mu \underset{FSD}{\succcurlyeq} (r_p \circ \pi_{\text{ref}})_\sharp \mu. \tag{5}$$

## 3   AOT: Alignment via Optimal Transport a Convex Relaxation Approach

Note that the paired and unpaired distributional preference constraints in Definitions 1 and 2 can be used in LLM alignment as follows:

$$\text{Find } \pi_\theta \in \mathcal{H} \text{ such that } (r_u \circ \pi_\theta)_\sharp \mu_+ \underset{\text{FSD}}{\succcurlyeq} (r_u \circ \pi_\theta)_\sharp \mu_- \qquad \text{(FSD unpaired)}$$

and

$$\text{Find } \pi_\theta \in \mathcal{H} \text{ such that } (r_p \circ \pi_\theta)_\sharp \mu \underset{\text{FSD}}{\succcurlyeq} (r_p \circ \pi_{\text{ref}})_\sharp \mu \qquad \text{(FSD paired)}$$

where $r_u$ are $r_p$ are given in Definitions 1 and 2 respectively, and $\mathcal{H}$ is a hypothesis class. Those two problems are instances of learning with stochastic orders introduced in [Domingo-Enrich et al., 2022], but in a simpler setting since the constraints are on one-dimensional distributions and the order considered is the first order rather than the convex order as considered in [Domingo-Enrich et al., 2022]. Note that both problems are special cases of the following generic optimization problem:

$$\text{Find } \theta \in \Theta \text{ such that } : U_\theta \underset{\text{FSD}}{\succcurlyeq} V_\theta \tag{6}$$

where $U_\theta$ and $V_\theta$ are real-valued random variables whose distributions depend on a parameter vector $\theta \in \Theta$. Note that for our FSD paired setting, $V_\theta = V$ (independent of $\theta$). Let $\mu_{U_\theta}$ and $\mu_{V_\theta}$ be the probability measures of $U_\theta$ and $V_\theta$ resp.

By the definition of FSD in Equation (1) we have:

$$U_\theta \underset{\text{FSD}}{\succcurlyeq} V_\theta \iff Q_{U_\theta}(t) \geq Q_{V_\theta}(t), \forall t \in [0, 1].$$

We can relax this problem to the following minimization problem:

$$\min_{\theta \in \Theta} \varepsilon(\theta) := \int_0^1 h(Q_{U_\theta}(t) - Q_{V_\theta}(t))dt, \tag{7}$$

where $h$ is a function penalizing each quantile's violation of FSD. The objective function (7) seeks to measure the violation of FSD, so that it can be minimized or eliminated. For instance, with $h$ the 0/1 loss (here $\mathbb{1}$ is the indicator function):

$$\min_{\theta \in \Theta} \int_0^1 \mathbb{1}_{Q_{U_\theta}(t) < Q_{V_\theta}(t)} dt, \tag{8}$$

This loss reminds us the misclassification 0/1 loss. Following classical convex relaxation of 0/1 losses in binary classification [Bartlett et al., 2006], we consider surrogates $h$ of the indicator function. Our choices for $h$ are motivated by the "almost-FSD" notions in the literature (See Appendix F for a discussion). In practice, we use smooth convex approximations of the 0/1 loss ($\mathbb{1}_{x<0}$) [Bartlett et al., 2006], for example for a margin $\beta > 0$ $h(x) = (\beta - x)_+^2$ the $\beta-$ **squared hinge loss** or $h(x) = \log(1 + \exp(-\beta x))$ the $\beta$-**logistic loss**. Although not a convex relaxation of the 0/1 loss, the least squares loss has been used in classification [Rosasco et al., 2004], and in the context of alignment, it was used in IPO [Azar et al., 2024] hence we use also $h(x) = (\beta - x)^2$, and refer to it as $\beta$-**Least Squares**. Further discussion of tradeoffs and benefits of different losses is in Appendix F, and formal assumptions on $h$ needed for the statistical theory are given in Assumption 1.

The cost function in (7) is still computationally challenging, if we were to solve the problem via gradient descent on $\theta$ this would require us to differentiate through the quantile operation. The following theorem from Santambrogio [2015] will be instrumental for us to cast the loss in (7) as an optimal transport problem with a convex cost $h$:

**Theorem 1** (Theorem 2.9 and Proposition 2.17 in Santambrogio [2015]). *Let $h : \mathbb{R} \to \mathbb{R}^+$ be a convex function we have for two real random variables $U, V$, with measures $\mu_U, \mu_V$:*

$$\int_0^1 h(Q_U(t) - Q_V(t))dt = \min_{\gamma \in \Pi(\mu_U, \mu_V)} \int h(u - v)d\gamma(u, v) = \mathsf{OT}_h(\mu_U, \mu_V)$$

*and $\gamma^* = (Q_U, Q_V)_\sharp \mathcal{L}_1([0, 1])$ is a minimizer (where $\mathcal{L}_1$ is the Lebesgue measure on $[0, 1]$ ). If furthermore $h$ is strictly convex $\gamma^*$ is the unique minimizer.*

Thanks to Theorem 1 we can write the problem (7), in the following equivalent form that we call Alignment via Optimal Transport (AOT) :

$$\min_{\theta \in \Theta} \int_0^1 h(Q_{U_\theta}(t) - Q_{V_\theta}(t))dt = \min_{\theta \in \Theta} \mathsf{OT}_h(\mu_{U_\theta}, \mu_{V_\theta}) = \min_{\theta \in \Theta} \min_{\gamma \in \Pi(\mu_{U_\theta}, \mu_{V_\theta})} \int h(u - v)d\gamma(u, v). \tag{9}$$

This formulation reveals that we have turned the stochastic dominance constraint to an inner one-dimensional optimal transport problem with a convex cost $h$. $\mathsf{OT}_h(\mu_{U_\theta}, \mu_{V_\theta})$ can be thought as a soft measure of the violation of the stochastic dominance of $U_\theta$ on $V_\theta$, hence by minimizing it as function of $\theta$ we are ensuring the optimal $\theta^*$ results in $U_{\theta^*}$ dominating $V_{\theta^*}$. Such OT problems with a smooth cost have been subject to theoretical and statistical study in one dimension as well as in high dimensions. For instance, [Manole and Niles-Weed, 2024] considered smooth convex costs, and [Hundrieser et al., 2022] considered more general smooth costs. [Groppe and Hundrieser, 2023] considered entropic regularization of optimal transport with general smooth costs.

**Computational Algorithm via Sorting** We consider here empirical measures and turn to solve the inner problem for a fixed $\theta$. We omit $\theta$ in what follows to simplify notation. We are interested in $\mathsf{OT}_h(\hat{\mu}_U, \hat{\mu}_V)$ where $\hat{\mu}_U = \frac{1}{n} \sum_{i=1}^n \delta_{u_i}$ $\hat{\mu}_V = \frac{1}{n} \sum_{i=1}^n \delta_{v_i}$. Given the convexity of $h$ and thanks to Theorem (1), the optimal coupling of $\mathsf{OT}_h(\hat{\mu}_{U_\theta}, \hat{\mu}_{V_\theta})$ is given by the north-west corner solution [Peyré and Cuturi, 2019] (Chapter 3, Section 3.4.2) that informally matches the $i-$th smallest element of $U$ with the $i-$th smallest element from $V$. More formally, if we sort the variables $u_i$ and get the order statistics (from min to max) $u^{(1)} \leq ... \leq u^{(n)}$ and same for $v_i$: $v^{(1)} \leq ... \leq v^{(n)}$. We have:

$$\mathsf{OT}_h(\hat{\mu}_U, \hat{\mu}_V) = \frac{1}{n} \sum_{i=1}^{n} h(u^{(i)} - v^{(i)}). \tag{10}$$

Back to (9), given empirical samples $\hat{\mu}_{U_\theta} = \frac{1}{n} \sum_{i=1}^{n} \delta_{u_\theta^i}$ and $\hat{\mu}_{V_\theta} = \frac{1}{n} \sum_{i=1}^{n} \delta_{v_\theta^i}$, let $u_\theta^{(i)}, v_\theta^{(i)}$ be the order statistics as function of $\theta$. We have therefore:

$$\min_{\theta \in \Theta} \mathsf{OT}_h(\hat{\mu}_{U_\theta}, \hat{\mu}_{V_\theta}) = \min_{\theta \in \Theta} \frac{1}{n} \sum_{i=1}^{n} h(u_\theta^{(i)} - v_\theta^{(i)}) \quad \text{(AOT)} \tag{11}$$

In Appendix G, we show that the gradients of the objective (11) are asymptotically unbiased for bounded distributions (see the statement for all conditions). Note that the sorting operation in (11) is a 1-Lipschitz function with discontinuous Jacobian.[2] Like the ReLU activation function, it can be easily optimized by gradient descent [Anil et al., 2019] (compare also sliced Wasserstein GANs). In practice, computing the gradient at any given step is done by first running the sorting algorithm and taking the gradient with respect to $\theta$ with the current assignment held fixed.

AOT **for Unpaired Preference**  Let $\hat{\mu}_+^n = \frac{1}{n} \sum_{i=1}^{n} \delta_{(x_{i,+}, y_{i,+})}$ and $\hat{\mu}_-^n = \frac{1}{n} \sum_{i=1}^{n} \delta_{(x_{i,-}, y_{i,-})}$. Our convex relaxation approach for unpaired FSD alignment given in (FSD unpaired) can therefore be cast as an AOT problem (given in Equation (11)) for

$$u_\theta^i = \log \frac{\pi_\theta(y_{i,+}|x_{i,+})}{\pi_{\mathrm{ref}}(y_{i,+}|x_{i,+})}, \quad v_\theta^i = \log \frac{\pi_\theta(y_{i,-}|x_{i,-})}{\pi_{\mathrm{ref}}(y_{i,-}|x_{i,-})}, \quad i = 1, \dots, n.$$

AOT **for Paired Preference**  Let $\hat{\mu}^n = \frac{1}{n} \sum_{i=1}^{n} \delta_{(x_i, y_{i,+}, y_{i,-})}$ be a paired preference empirical measure. Our convex relaxation approach for paired FSD alignment given in (FSD paired) can be there cast as an AOT problem (given in Equation (11)) for:

$$u_\theta^i = \log \frac{\pi_\theta(y_{i,+}|x_i)}{\pi_\theta(y_{i,-}|x_i)}, \quad v_\theta^i = \log \frac{\pi_{\mathrm{ref}}(y_{i,+}|x_i)}{\pi_{\mathrm{ref}}(y_{i,-}|x_i)}, \quad i = 1, \dots, n.$$

AOT **with Soft Sorting**  One caveat of the alternating optimization for AOT between $\theta$ and solving the inner optimal transport problem with hard sorting is that the gradient with respect to the parameter $\theta$ for fixed permutations has dependency in $\theta$ on the order statistics level only and not through the sorting routine. To alleviate that, we propose to use SoftSorting [Blondel et al., 2020, Cuturi et al., 2019] that uses an entropic regularization to find a smoothed permutations via a Sinkhorn algorithm, which in turn allows the back-propagation on $\theta$ to depend not only via the order statistics but also via the computational graph of SoftSorting.

Algorithms 1 and 2 in Appendix B summarize our AOT approach for distributional preference alignment in the unpaired and paired setting.

## 4 Statistical Analysis

In this section, we focus on the statistical analysis of unpaired-AOT and defer paired-AOT to Appendix E since it has a similar analysis. We make the following assumptions on the OT cost $h$, the reward $r$, and the policy hypothesis class $\mathcal{H}$.

**Assumption 1** (OT cost). *Let $M, R > 0$ be finite positive constants. We assume that the loss $h : [-M, M] \to [0, R]$, is convex $L$-Lipschitz and bounded. $h$ is a convex function (E.g. a relaxation of the 0/1 loss such that $h(t) > h(t')$, for $t < 0$ and $t' > 0$).*

**Assumption 2** (Reward). *We assume that $r$ is bounded so that $r \circ \pi_\theta(x, y) \in [-M, M]$.*

**Assumption 3** (Assumption on the hypothesis class of the policy). *We assume $\pi_{\mathrm{ref}}, \pi_\theta \in \mathcal{H} = \{\pi_\theta : \text{such that } r \circ \pi_\theta \text{ differentiable in } \theta \text{ and } \sup_{x \in \mathcal{X}, y \in \mathcal{Y}} \|\nabla_\theta r \circ \pi_\theta(y|x)\| \leq L', \theta \in \Theta \subset B_2(r_0, d_\theta)\}$, for $L', r_0 > 0$. .*

---

[2]Its Jacobian is a permutation matrix at every point except a measure-zero set where it is not differentiable.

**Assumption 4.** *There exists $\pi_\theta \in \mathcal{H}$ such that $(r \circ \pi_\theta)_\sharp \mu_+ \underset{FSD}{\succcurlyeq} (r \circ \pi_\theta)_\sharp \mu_-$.*

Assumption 1 is satisfied for example by the hinge squared loss $h(t) = (-t)_+^2$ by the logistic loss $h(t) = \log(1 + e^{-\beta t})$, for $t \in [-M, M]$. Assumption 2 on the boundedness of the rewards can be imposed by clamping the values of the logits of the policies to $[-M, M]$, which is common practice in practical implementations of LLM alignment. Assumption 3 is a technical assumption needed to control the covering number of the $r \circ \mathcal{H}$. Assumption 4 ensures the existence of the minimizer in $\mathcal{H}$. We overload notations in what follows and refer to $r_u$ and $r_p$ as $r$ to simplify the presentation. By our relaxation approach described in Section 3 we can relax the unpaired stochastic dominance constraint problem given in (FSD unpaired) to:

$$\min_{\pi_\theta \in \mathcal{H}} \int_0^1 h\left(Q_{(r \circ \pi_\theta)_\sharp \mu_+}(t) - Q_{(r \circ \pi_\theta)_\sharp \mu_-}(t)\right) dt = \min_{\pi_\theta \in \mathcal{H}} \mathsf{OT}_h\left((r \circ \pi_\theta)_\sharp \mu_+, (r \circ \pi_\theta)_\sharp \mu_-\right)$$

$$(\text{uAOT}_h)$$

Define the OT cost $c : [-M, M] \times [-M, M] \to [0, R]$ such that $c(z, z') = h(z - z')$, for $z, z \in [-M, M]$. Define the $c$-transform of a function $\varphi : [-M, M] \to \mathbb{R}$:

$$\varphi^c(z) = \inf_{z' \in [-M, M]} h(z - z') - \varphi(z).$$

In our setting, a function is called $c$-concave if there exists $\psi : [-M, M] \to \mathbb{R}$ such that $\varphi = \psi^c$. Define:

$$\mathcal{F}_c = \{\varphi : [-M, M] \to [-R, R], \varphi \text{ is } c\text{-concave, with } \|\varphi^c\|_\infty \leq R\}$$

By duality (Theorem 5.10 in [Villani, 2009]) we have:

$$\mathsf{OT}_h\left((r \circ \pi_\theta)_\sharp \mu_+, (r \circ \pi_\theta)_\sharp \mu_-\right) = \sup_{\varphi \in \mathcal{F}_c} \int \varphi(r \circ \pi_\theta) d\mu_+ - \int \varphi^c(r \circ \pi_\theta) d\mu_-.$$

Replacing the dual expression of $\mathsf{OT}_h$ in (uAOT$_h$), we see that (uAOT$_h$) can be cast as a min-max problem:

$$\min_{\pi_\theta \in \mathcal{H}} \sup_{\varphi \in \mathcal{F}_c} \int \varphi(r \circ \pi_\theta) d\mu_+ - \int \varphi^c(r \circ \pi_\theta) d\mu_-. \tag{12}$$

Given samples $\hat{\mu}_+^n = \frac{1}{n} \sum_{i=1}^n \delta_{(x_{i,+}, y_{i,+})}$ and $\hat{\mu}_-^n = \frac{1}{n} \sum_{i=1}^n \delta_{(x_{i,-}, y_{i,-})}$, the empirical problem is:

$$\min_{\pi_\theta \in \mathcal{H}} \sup_{\varphi \in \mathcal{F}_c} \int \varphi(r \circ \pi_\theta) d\hat{\mu}_+^n - \int \varphi^c(r \circ \pi_\theta) d\hat{\mu}_-^n. \tag{13}$$

Recall that $\mathsf{OT}_h$ is a measure of the violation of stochastic dominance of $(r \circ \pi_\theta)_\sharp \mu_+$ on $(r \circ \pi_\theta)_\sharp \mu_-$. We have the following result on the sample complexity of the violation of stochastic dominance:

**Theorem 2** (Sample Complexity of Dominance Violation for AOT Unpaired). *Let $\pi_{\theta^*}$ be the population minimizer of (uAOT$_h$) and $\pi_{\hat{\theta}_n}$ be the solution of the empirical problem (13). We have the following sample complexity bound for the violation of stochastic dominance in* AOT *unpaired:*

$$\mathbb{E}\, \mathsf{OT}_h\left((r \circ \pi_{\hat{\theta}_n})_\sharp \mu_+, (r \circ \pi_{\hat{\theta}_n})_\sharp \mu_-\right) \leq \underbrace{\mathsf{OT}_h\left((r \circ \pi_{\theta^*})_\sharp \mu_+, (r \circ \pi_{\theta^*})_\sharp \mu_-\right)}_{\textit{Optimal Almost FSD Violation}}$$

$$+ \underbrace{2\mathcal{R}_n(\mathcal{F}_c; (r \circ \pi_{\theta^*})_\sharp \mu_+) + 2\mathcal{R}_n(\mathcal{F}_c^c; (r \circ \pi_{\theta^*})_\sharp \mu_-)}_{\textit{One dimensional OT sample complexity with optimal } \theta^*} + \underbrace{2\mathcal{R}_n(\mathcal{F}_c \circ r \circ \mathcal{H}; \mu_+) + 2\mathcal{R}_n(\mathcal{F}_c^c \circ r \circ \mathcal{H}; \mu_-)}_{\textit{Complexity of learning in } \mathcal{H} \textit{ via the 1D OT problem}},$$

*where $\mathcal{R}_n(\mathcal{F}; \nu) = \mathbb{E} \sup_{\varphi \in \mathcal{F}} \left|\frac{1}{n} \sum_{i=1}^n \sigma_i \varphi(Z_i)\right|$ is the Rademacher Complexity and for $i = 1 \ldots n$, $\sigma_i$ are independent Rademacher random variables and $Z_i \sim \nu$ iid.*

By considering our assumptions on the cost, the reward, and the hypothesis class, we obtain the parametric rate in $n$:

**Corollary 1.** *(Informal) Under Assumptions 1, 2 and 3 we have:*

1. $\mathbb{E}\, \mathsf{OT}_h\left((r \circ \pi_{\hat{\theta}_n})_\sharp \mu_+, (r \circ \pi_{\hat{\theta}_n})_\sharp \mu_-\right) - \mathsf{OT}_h\left((r \circ \pi_{\theta^*})_\sharp \mu_+, (r \circ \pi_{\theta^*})_\sharp \mu_-\right) \lesssim n^{-\frac{1}{2}}$, *where* $\lesssim$ *refers to inequality up to constants that depend only on constants in the assumptions.*

2. If in addition Assumption 4 holds and $h(t) = (-t)_+^2$, we have:
$$\mathbb{E}\, \mathsf{OT}_h \left( (r \circ \pi_{\hat{\theta}_n})_\sharp \mu_+, (r \circ \pi_{\hat{\theta}_n})_\sharp \mu_- \right) \lesssim n^{-\frac{1}{2}}.$$

We see that under our assumptions and for the hinge loss squared, the expected violation of the desired dominance in AOT unpaired converges to zero as $n \to \infty$.

**Remark 1.** *While in Section 3, we used the primal formulation to compute $\mathsf{OT}_h$ due to its computational appeal thanks to the sorting algorithm we used for analyzing the sample complexity the dual of $\mathsf{OT}_h$. The dual reveals the game theoretic aspect of $\mathsf{AOT}$ as a min-max game between the policy $\pi_\theta$ and the dual potential $\varphi_c$ that imposes FSD on the preference we want to infuse to the policy.*

## 5 Experiments

In this section, we evaluate the performance of the proposed AOT method on a diverse set of base LLMs and datasets, comparing with currently available alternative alignment algorithms.

**LLM Alignment Alternatives** We compared AOT with current state-of-the-art alignment approaches, specifically Direct Preference Optimization (DPO) [Rafailov et al., 2024], Kahneman-Tversky Optimization (KTO) [Ethayarajh et al., 2024] and Identity Policy Optimization (IPO) [Azar et al., 2024]. DPO and IPO operate on paired preference data, while KTO can handle both paired and unpaired prompt/response samples.

|  | AlpacaEval (GPT4) | ARC | Hellaswag | MMLU | Truthful | Winogrande | GSM8K |
|---|---|---|---|---|---|---|---|
| AOT paired | 29.9 | 82.5 | 66.1 | 62.9 | 50.8 | 74.4 | 53.1 |
| AOT unpaired | **31.3** | 82.5 | **66.2** | 62.8 | **51.1** | 74.4 | 51.8 |
| DPO | 27.4 | **82.8** | 65.8 | **63.1** | 50.6 | 74.3 | 52.0 |
| KTO | 24.9 | 82.7 | 65.4 | 63.0 | 48.7 | **74.9** | **53.9** |
| IPO | 27.7 | 82.4 | 65.1 | 63.0 | 46.5 | 74.0 | 52.3 |
| Merlinite-7B | 17.1 | 81.6 | 63.2 | 62.6 | 42.0 | 73.9 | 45.2 |

Table 1: `Merlinite-7B` trained on UltraFeedback Binarized. AOT results in the best performing LLM as compared to the alternative alignment algorithms on AlpacaEval, and is competitive across the other benchmarks that are evaluated in the zero shot regime.

**Reference Models** Traditionally, model alignment is the third and final step applied to the LLM that already has gone through original pretraining and supervised fine-tuning. For our experiments, we selected a range of models at various stages and with different levels of performance, all in the family of 7B-parameter models. Specifically, we used `Merlinite-7B` [Sudalairaj et al., 2024], which is a variant of `Mistral-7B-v0.1` that has been instruction-tuned (SFT) on data from a synthetic data generator using a taxonomy-driven data curation process. In Appendix H we also cover other popular LLMs, such as `Mistral-7B` [Jiang et al., 2023], `OpenHermes-2.5-Mistral-7B` [Teknium, 2024], `Starling` [Zhu et al., 2023], `Mistral-7B` Jiang et al. [2023], and `Llama3-8B` [AI@Meta, 2024].
**Datasets** For our experiments, we used both paired and unpaired datasets. For the paired dataset, we used the UltraFeedback binarized dataset from [Tunstall et al., 2023b], containing over 60K training samples, where for each prompt, there is a pair of chosen (preferred) and rejected (not preferred) responses. This alignment dataset is widely used, and all compared alignment techniques are well-suited for it. For unpaired datasets, we used PKU BeaverTails [Ji et al., 2023] with over 300K samples and HelpSteer [Wang et al., 2023] with around 35K samples. Here, for each prompt, there is only a single response with a score defined by some attributes (e.g., safety, faithfulness, helpfulness, etc.). We used the sum of attribute values and thresholded by the median to binarize the responses into chosen and rejected. For this unpaired dataset, only KTO and our AOT are applicable.

**Metrics** To measure the performance of different alignment methods, we used popular evaluation metrics, AlpacaEval [Dubois et al., 2024] and Open LLM benchmark [Beeching et al., 2023]. We note that Alpaca uses `GPT4` model as a judge to compare candidate responses to `GPT4`-based references on a set of 805 challenging questions. The `GPT4`-based evaluations are expensive, so to limit our expenses, we also employed a very strong and capable `Llama3-70B-Instruct` [AI@Meta, 2024] as a judge. As we show in Appendix H in Table 2, the order determined by `Llama3-70B-Instruct`

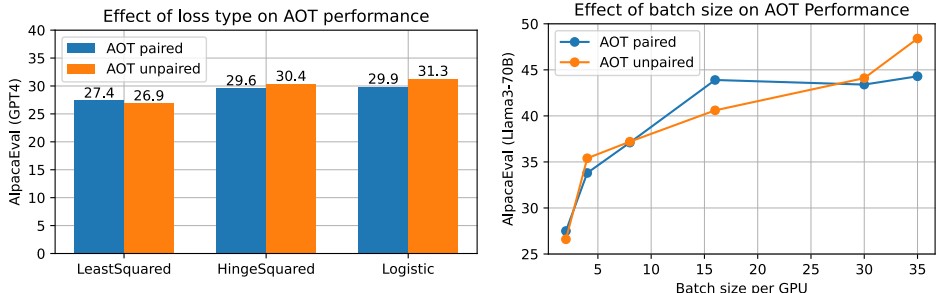

Figure 2: Impact of batch size and loss type on AOT performance. The batch size is the effective number of samples in the mini-batch per GPU. We found the logistic loss to be performing better than least squared or hinge squared losses (all using $\beta = 0.01$). As we increase batch size, we also observed improvement in AOT performance, which is expected as more samples per minibatch results in a better effect of stochastic dominance (conforming Corollary 1).

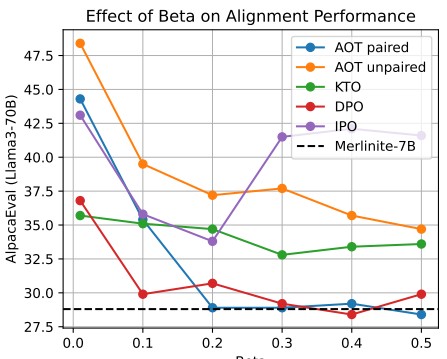

Figure 3: Impact of ($\beta$) parameter on performance of different alignment algorithms. $\beta$ controls the divergence of the policy model from the initial reference model (low beta - more divergence, high beta - less divergence). We see a general trend that with higher betas, LLMs alignment decreases the performance. Hence, for all experiments, we selected $\beta = 0.01$ as a default value.

and GPT4 is the same (the absolute score values are different), providing a better free alternative LLM-judge for local Alpaca evaluations. For intermediate results, we also employed Tiny Benchmarks [Maia Polo et al., 2024] to approximate original metrics and provide fast feedback during the initial development. We also evaluated the aligned models on six key benchmarks from the Open LLM Leaderboard: AI2 Reasoning Challenge - ARC (grade-school science questions), HellaSwag (commonsense inference), MMLU (multi-task accuracy), TruthfulQA (tendency to reproduce falsehoods), Winogrande (commonsense reasoning) and GSM8K (grade school math word problems). Note that in all the above benchmarks we use 0-shot prompts, a more challenging setting as opposed to commonly used few-shot prompting.

**Experimental Setup** Our implementation is based on the HuggingFace Alignment Handbook Tunstall et al. [2023a]. As we show in Appendix in Section B, the changes needed to adapt HF TRL trainer [von Werra et al., 2020] for AOT are minimal and therefore can easily be adapted by the community. For each run our compute setup consisted of 8 H100 GPUs. We used LoRA [Hu et al., 2021] for parameter-efficient fine-tuning during alignment and the FSDP (Fully-Sharded Data-Parallel) setup to train the model over multiple GPUs. Under this setup, the training of each 7B-parameter model on the UltraFeedback dataset took approximately one hour. The evaluation on AlpacaEval and Open LLM benchmarks took one additional hour to get the final results.

**Results** In Table 1, we present the main results of comparing AOT to other baselines (KTO, DPO, and IPO) on paired UltraFeedback binarized dataset. On AlpacaEval (GPT4), our AOT unpaired approach scores 31.3%, which is a significant gain from the base Merlinite-7B model. As of time of this writing (May 22nd, 2024), this result places our AOT aligned LLM on AlpacaEval LeaderBoard ahead of such strong competitors as KTO-Mistral-PAIR [Ethayarajh et al., 2023] and other 7B-parameter models, reaching the level of Mixtral-8x22B-v0.1 (see Figure 4 in Appendix for an illustration). On other LLM benchmarks AOT performs competitively to other baselines. As mentioned earlier, these evaluations are done using 0-shot prompts, leading to a more challenging setting and resulting in overall lower performance across metrics and baselines. For other base LLMs we show their performance in Appendix H (see Tables 3, 4, 5, and 6).

We also examined the effect of batch size and the choice of loss function on AOT performance, results shown in Fig. 2. As the batch size increases, AlpacaEval (based on `Llama3-70B-instruct`) also increases in line with our theory in Corollary 1. Note that our current setup (FSDP over 8 H100 GPUs) limits our batch size to 35 samples per GPU. We have also examined the impact of beta (controlling divergence of policy from reference) on AOT performance in Fig. 3. We noticed a trend that with higher betas the performance of LLMs alignment decreases, thus we set $\beta = 0.01$. Ablation results comparing hard and soft sorting as well as the variance of AlpacaEval scores across multiple runs in Appendix H (Tables 7 and 10) show the overall robustness of AOT .

## 6    Conclusion

We present in this paper Distributional Alignment via Optimal Transport (AOT) for large language models. The AOT cost can be cast as a one-dimensional optimal transport problem with a smooth and convex cost that penalizes violations of the dominance of the chosen on rejected marginals. AOT enjoys parametric statistical rates. We showed with extensive experimentation on various paired and unpaired datasets, base models, and different loss functions, that AOT alignment robustly leads to aligned models that outperform alternative alignment strategies such as DPO, KTO and IPO on the Alpaca Benchmark, leading to the best 7B model to date on that benchmark as of the time of writing. On other benchmarks such as the open LLM leaderboard AOT leads to competitive results.

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

# A   Broader Impact and Limitations

In this paper, we introduced an alignment method for LLMs that can capture rewards at the distributional level without the requirement of paired preference data. Our algorithm was derived by imposing the stochastic dominance of positive reward distribution over negative distributions through an Optimal Transport formulation. It enjoys a very simple algorithmic implementation in terms of a closed-form expression via sorting on empirical measures. Empirically, our algorithm demonstrates excellent results by allowing us to train 7B parameter models that achieve state-of-the-art evaluation results on Open LLM Benchmarks and AlpacaEval.

In terms of broader societal impact, we would like to highlight the benefits that our AOT algorithm will bring to RLHF by enabling a more robust distributional alignment of LLMs, improving their ability to follow instructions accurately, and aligning their responses with human values.

Our work shares the same limitations and possible negative broader societal impacts as the majority of RLHF work. The algorithm is fundamentally limited by the training dataset used for alignment and might, therefore, contribute to amplifying various types of bias present in the data. In addition, alignment through AOT is not enough to address aspects related to the security and safety of LLM deployment. In general, better performance on a given set of benchmarks following alignment does not imply better performance across the board in other tasks, and ad-hoc evaluation specific to each task of interest is warranted.

# B Algorithms and Pytorch Code In Hugging Face TRL

**Algorithm 1** AOT Unpaired

1: **Input:** $\pi_\theta, \pi_{\text{ref}}, \beta > 0, \varepsilon > 0,$
2: **Unpaired Preference Data:** "Chosen" $\hat{\mu}_+^n = \frac{1}{n}\sum_{i=1}^n \delta_{(x_{i,+}, y_{i,+})}$ and "Rejected"
3: $\hat{\mu}_-^n = \frac{1}{n}\sum_{i=1}^n \delta_{(x_{i,-}, y_{i,-})}.$
4: **for** iter $\leftarrow 1, n_{\text{iter}}$ **do**
5:     **Get a Positive/Negative mini-batch**
6:     $\{(x_{i,+}, y_{i,+}) \sim \hat{\mu}_+^n, i = 1 \ldots b\}$
7:     $\{(x_{i,-}, y_{i,-}) \sim \hat{\mu}_-^n, i = 1 \ldots b\}$
8:     **Compute Rewards for** $i = 1 \ldots b$
9:     $u_\theta^i = \log \frac{\pi_\theta(y_{i,+}|x_{i,+})}{\pi_{\text{ref}}(y_{i,+}|x_{i,+})}$
10:    $v_\theta^i = \log \frac{\pi_\theta(y_{i,-}|x_{i,-})}{\pi_{\text{ref}}(y_{i,-}|x_{i,-})}$
11:    **Sort Rewards**
12:    **if** hard sort **then**
13:       $(u^{(1)} \ldots u^{(b)}) = \text{Sort}(u_\theta^i)$
14:       $(v^{(1)} \ldots v^{(b)}) = \text{Sort}(v_\theta^i)$
15:    **else if** soft sort **then**
16:       $(u^{(1)} \ldots u^{(b)}) = \text{SoftSort}(u_\theta^i, \varepsilon)$
17:       $(v^{(1)} \ldots v^{(b)}) = \text{SoftSort}(v_\theta^i, \varepsilon)$
18:    **end if**
19:    **Compute AOT logistic loss**
20:    $\ell_\theta = -\frac{1}{b}\sum_{i=1}^b \log \sigma(\beta(u_\theta^{(i)} - v_\theta^{(i)}))$
21:    **Update** $\theta$
22:    $\theta \leftarrow \text{PagedAdamw32bit}(\nabla_\theta \ell(\theta))$
23: **end for**
24: **Return** $\pi_\theta$

**Algorithm 2** AOT Paired

1: **Input:** $\pi_\theta, \pi_{\text{ref}}, \beta > 0, \varepsilon > 0,$
2: **Paired Preference Data:** $\hat{\mu}^n = \frac{1}{n}\sum_{i=1}^n \delta_{(x_i, y_{i,+}, y_{i,-})}$
3: **for** iter $\leftarrow 1, n_{\text{iter}}$ **do**
4:     **Get a Positive/Negative mini-batch**
5:     $\{(x_i, y_{i,+}, y_{i,-}) \sim \hat{\mu}^n, i = 1 \ldots b\}$
6:     **Compute Margins for** $i = 1 \ldots b$
7:     $u_\theta^i = \log \frac{\pi_\theta(y_{i,+}|x_i)}{\pi_\theta(y_{i,-}|x_i)}$
8:     $v_\theta^i = \log \frac{\pi_{\text{ref}}(y_{i,+}|x_i)}{\pi_{\text{ref}}(y_{i,-}|x_i)}$
9:     **Sort Margins**
10:    **if** hard sort **then**
11:       $(u^{(1)} \ldots u^{(b)}) = \text{Sort}(u_\theta^i)$
12:       $(v^{(1)} \ldots v^{(b)}) = \text{Sort}(v_\theta^i)$
13:    **else if** soft sort **then**
14:       $(u^{(1)} \ldots u^{(b)}) = \text{SoftSort}(u_\theta^i, \varepsilon)$
15:       $(v^{(1)} \ldots v^{(b)}) = \text{SoftSort}(v_\theta^i, \varepsilon)$
16:    **end if**
17:    **Compute AOT logistic loss**
18:    $\ell_\theta = -\frac{1}{b}\sum_{i=1}^b \log \sigma(\beta(u_\theta^{(i)} - v_\theta^{(i)}))$
19:    **Update** $\theta$
20:    $\theta \leftarrow \text{PagedAdamw32bit}(\nabla_\theta \ell(\theta))$
21: **end for**
22: **Return** $\pi_\theta$

```python
import torch
import torchsort
```

```python
def dpo_loss( ...
...
elif self.loss_type == "AOT_unpair":
    chosen_logratios = policy_chosen_logps - reference_chosen_logps
    rejected_logratios = policy_rejected_logps - reference_rejected_logps
    if self.sort_type == "hard_sort":
        chosen_logratios_sorted , _ = torch.sort(chosen_logratios , dim=0)
        rejected_logratios_sorted , _ = torch.sort(rejected_logratios , dim=0)
    elif self.sort_type == "soft_sort":
        chosen_logratios_sorted = torchsort.soft_sort(chosen_logratios , regularization_strength =0.1)
        rejected_logratios_sorted = torchsort.soft_sort(rejected_logratios , regularization_strength
            =0.1)
    delta_sorted = chosen_logratios_sorted - rejected_logratios_sorted
    if self.AOT_loss == "hinge":
        losses = torch.relu(self.beta - delta_sorted)**2
    elif self.AOT_loss == "logistic":
        losses = (
            -F.logsigmoid(self.beta *delta_sorted) * (1 - self.label_smoothing)
            - F.logsigmoid(-self.beta * delta_sorted) * self.label_smoothing
            )
```

```python
elif self.loss_type == "AOT_pair":
    pi_logratios = policy_chosen_logps - policy_rejected_logps
    ref_logratios = reference_chosen_logps - reference_rejected_logps
    if self.sort_type == "hard_sort":
        pi_logratios_sorted , _ =  torch.sort(pi_logratios , dim=0)
        ref_logratios_sorted , _ =  torch.sort(ref_logratios , dim=0)
    elif self.sort_type == "soft_sort":
        pi_logratios_sorted = torchsort.soft_sort(pi_logratios , regularization_strength =0.1)
        ref_logratios_sorted = torchsort.soft_sort(ref_logratios , regularization_strength =0.1)
    delta_sorted = pi_logratios_sorted - ref_logratios_sorted
    if self.AOT_loss == "hinge":
        losses = torch.relu(self.beta - delta_sorted)**2
    elif self.AOT_loss == "logistic":
        losses = (
            -F.logsigmoid(self.beta *delta_sorted) * (1 - self.label_smoothing)
            - F.logsigmoid(-self.beta * delta_sorted) * self.label_smoothing
            )
```

## C  Proofs

*Proof of Theorem 2.* Let

$$\varepsilon(\theta, \mu_+, \mu_-) = \sup_{\varphi \in \mathcal{F}_c} \int \varphi(r \circ \pi_\theta) d\mu_+ - \int \varphi^c(r \circ \pi_\theta) d\mu_-$$

where for $z, z' \in \mathbb{R}$ :

$$\varphi^c(z) = \inf_{z' \in [-M, M]} h(z - z') - \varphi(z)$$

and (a function $f$ is $c-$ concave if there exits $g$ such that $f = g^c$).

The population problem :

$$\min_{\pi_\theta \in \mathcal{H}} \varepsilon(\theta, \mu_+, \mu_-) \tag{14}$$

Given samples $\hat{\mu}_+^n = \frac{1}{n} \sum_{i=1}^n \delta_{(x_+^i, y_+^i)}$ and $\hat{\mu}_-^m = \frac{1}{m} \sum_{i=1}^m \delta_{(x_-^i, y_-^i)}$, the empirical problem is :

$$\varepsilon(\theta, \hat{\mu}_+^n, \hat{\mu}_-^m) = \sup_{\varphi \in \mathcal{F}_c} \int \varphi(r \circ \pi_\theta) d\hat{\mu}_+^n - \int \varphi^c(r \circ \pi_\theta) d\hat{\mu}_-^m$$

and the ERM problem is :

$$\min_{\pi_\theta \in \mathcal{H}} \varepsilon(\theta, \hat{\mu}_+^n, \hat{\mu}_-^m)$$

Let $\hat{\theta}_{m,n}$ be the minimizer of the ERM we have for any $\theta$, by the definition of the minimizer:

$$\varepsilon(\hat{\theta}_{m,n}, \hat{\mu}_+^n, \hat{\mu}_-^m) \leq \sup_{\varphi \in \mathcal{F}_c} \int \varphi(r \circ \pi_\theta) d\hat{\mu}_+^n - \int \varphi^c(r \circ \pi_\theta) d\hat{\mu}_-^m$$

$$\leq \sup_{\varphi \in \mathcal{F}_c} \int \varphi(r \circ \pi_\theta) d\mu_+ - \int \varphi^c(r \circ \pi_\theta) d\mu_-$$

$$+ \sup_{\varphi \in \mathcal{F}_c} \int \varphi(r \circ \pi_\theta) d(\hat{\mu}_+^n - \mu_+)$$

$$+ \sup_{\varphi \in \mathcal{F}_c} \int \varphi^c(r \circ \pi_\theta) d(\mu_- - \hat{\mu}_-^m)$$

Let $\theta^*$ be the minimizer of (14) for $\theta = \theta^*$ in the above inequality, and taking expectations on the randomness of the samples we obtain

$$\mathbb{E}\varepsilon(\hat{\theta}_{m,n}, \hat{\mu}_+^n, \hat{\mu}_-^m) \leq \mathbb{E}\varepsilon(\theta^*, \mu_+ . \mu_-) + \mathbb{E} \sup_{\varphi \in \mathcal{F}_c} \int \varphi(r \circ \pi_{\theta^*}) d(\hat{\mu}_+^n - \mu_+)$$

$$+ \mathbb{E} \sup_{\varphi \in \mathcal{F}_c} \int \varphi^c(r \circ \pi_{\theta^*}) d(\mu_- - \hat{\mu}_-^m)$$

On the other hand by symmetrization we have:

$$\mathbb{E} \sup_{\varphi \in \mathcal{F}_c} \int \varphi(r \circ \pi_{\theta^*}) d(\hat{\mu}_+^n - \mu_+) \leq 2\mathcal{R}_n(\mathcal{F}_c), \tag{15}$$

where $\mathcal{R}_n(\mathcal{F}) = \mathbb{E} \sup_{\varphi \in \mathcal{F}} \left| \frac{1}{N} \sum_{i=1}^N \sigma_i \varphi(X_i) \right|$, $\sigma_i$ are independent rademacher random variables and $X_i \sim (r \circ \pi_{\theta^*})_\sharp \mu_+$ iid ($X_i \in \mathbb{R}$). and similarly we have:

$$\mathbb{E} \sup_{\varphi \in \mathcal{F}_c} \int \varphi^c(r \circ \pi_{\theta^*}) d(\mu_- - \hat{\mu}_-^m) \leq \mathbb{E} \sup_{\varphi^c \in \mathcal{F}_c^c} \int \varphi^c(r \circ \pi_{\theta^*}) d(\mu_- - \hat{\mu}_-^m) \leq 2\mathcal{R}_m(\mathcal{F}_c^c)$$

We have finally:

$$\mathbb{E}\varepsilon(\hat{\theta}_{m,n}, \hat{\mu}_+^n, \hat{\mu}_-^m) \leq \varepsilon(\theta^*, \mu_+ . \mu_-) + 2\mathcal{R}_n(\mathcal{F}_c) + 2\mathcal{R}_m(\mathcal{F}_c^c) \tag{16}$$

Turning now to :

$$\varepsilon(\hat{\theta}_{m,n}, \mu_+, \mu_-) = \sup_{\varphi \in \mathcal{F}_c} \int \varphi(r \circ \pi_{\hat{\theta}_{m,n}}) d\mu_+ - \int \varphi^c(r \circ \pi_{\hat{\theta}_{m,n}}) d\mu_-$$

$$\leq \varepsilon(\hat{\theta}_{m,n}, \hat{\mu}_+^n, \hat{\mu}_-^m) + \sup_{\varphi \in \mathcal{F}_c} \int \varphi(r \circ \pi_{\hat{\theta}_{m,n}}) d(\mu_+ - \mu_+^n)$$

$$+ \sup_{\varphi \in \mathcal{F}_c} \int \varphi^c(r \circ \pi_{\hat{\theta}_{m,n}}) d(\mu_-^m - \mu_-)$$

$$\leq \varepsilon(\hat{\theta}_{m,n}, \hat{\mu}_+^n, \hat{\mu}_-^m) + \sup_{\pi_\theta \in \mathcal{H}} \sup_{\varphi \in \mathcal{F}_c} \int \varphi(r \circ \pi_\theta) d(\mu_+ - \mu_+^n)$$

$$+ \sup_{\pi_\theta \in \mathcal{H}} \sup_{\varphi \in \mathcal{F}_c} \int \varphi^c(r \circ \pi_\theta) d(\mu_-^m - \mu_-)$$

Taking expectations we obtain:

$$\mathbb{E}\varepsilon(\hat{\theta}_{m,n}, \mu_+, \mu_-) \leq \mathbb{E}\varepsilon(\hat{\theta}_{m,n}, \hat{\mu}_+^n, \hat{\mu}_-^m) + \mathbb{E} \sup_{\pi_\theta \in \mathcal{H}} \sup_{\varphi \in \mathcal{F}_c} \int \varphi(r \circ \pi_\theta) d(\mu_+ - \mu_+^n)$$

$$+ \mathbb{E} \sup_{\pi_\theta \in \mathcal{H}} \sup_{\varphi^c \in (\mathcal{F}_c)^c} \int \varphi^c(r \circ \pi_\theta) d(\mu_-^m - \mu_-)$$

$$\leq \underbrace{\varepsilon(\theta^*, \mu_+.\mu_-)}_{\text{Optimal FSD Violation}} + \underbrace{2\mathcal{R}_n(\mathcal{F}_c) + 2\mathcal{R}_m(\mathcal{F}_c^c)}_{\text{One dimensional OT complexity with optimal } \theta^*}$$

$$+ \underbrace{2\mathcal{R}_n(\mathcal{F}_c \circ r \circ \mathcal{H}) + 2\mathcal{R}_m(\mathcal{F}_c^c \circ r \circ \mathcal{H})}_{\text{Complexity of learning in } \mathcal{H} \text{ via the 1D OT problem}},$$

where

$$\mathcal{R}_n(\mathcal{F}_c \circ r \circ \mathcal{H}) = \frac{1}{n} \mathbb{E} \sup_{\pi_\theta \in \mathcal{H}} \sup_{\varphi \in \mathcal{F}_c} \left| \sum_{i=1}^n \sigma_i \varphi(r \circ \pi_\theta(x_i^+, y_i^+)) \right| \tag{17}$$

$\square$

# D   Bounding Rademacher Complexities

*Proof of Corollary 1.* Define the uniform metric entropy of a class of real valued functions $\mathcal{F}$ on a set $\mathcal{X}$ as the logarithm of the covering number with respect to the uniform norm $\|.\|_\infty$, for $\varepsilon > 0$, this is defined as follows:

$$\mathcal{N}(\varepsilon, \mathcal{F}, \|.\|_\infty) := \inf \left\{ n \in \mathbb{N} \,\middle|\, \text{there exists } f_1 \dots f_n : \mathcal{X} \to \mathbb{R} \text{ with } \sup_{f \in \mathcal{F}} \min_{1 \leq i \leq n} \|f - f_i\|_\infty \leq \varepsilon \right\}$$

As observed in [Hundrieser et al., 2022] the c-transformation with bounded cost is a lipchitz operation under the uniform norm and since $f^{cc} = f$ we have by Lemma 2.1 in Munk :

$$\mathcal{N}(\varepsilon, \mathcal{F}_c^c, \|.\|_\infty) = \mathcal{N}(\varepsilon, \mathcal{F}_c, \|.\|_\infty) \tag{18}$$

Now turning to the Rademacher complexity of a class $\mathcal{F}$, it is dominated by Dudley's entropy integral (Theorem 16 in Luxburg and Bousquet ):

$$\mathcal{R}_n(\mathcal{F}) \leq \inf_{\delta \in [0,R]} \left( 2\delta + \sqrt{32} \frac{1}{\sqrt{n}} \int_{\delta/4}^R \sqrt{\log \mathcal{N}(\varepsilon, \mathcal{F}, \|.\|_\infty)} d\varepsilon \right) \tag{19}$$

Note that the cost we are using is $c(z, z') = h(z - z')$, for $z, z \in [-M, M]$. The domain on which the cost being a closed interval is convex and compact. By lipchitzity of $h$ (Assumption 1) and denoting $L$ its lipchitz constant, $c(, z')$ is lipchitz for all $z' \in [-M, M]$. Equivalently $c(, z')$ is $(\alpha, \Lambda)$ Hölder smooth , for $\alpha = 1$ and $\Lambda = L$, and hence our setup falls under the Assumptions of Theorem 3.11 in [Hundrieser et al., 2022] for Holder smooth costs defined on convex and compact sets and we have:

$$\log \mathcal{N}(\varepsilon, \mathcal{F}_c, \|.\|_\infty) \lesssim \varepsilon^{-\frac{d}{\alpha}}.$$

Hence in our case $\alpha = 1$ and $d = 1$ this leads to

$$\log \mathcal{N}(\varepsilon, \mathcal{F}_c, \|.\|_\infty) \lesssim \varepsilon^{-1}$$

Replacing this in Equation (19) we obtain: $\mathcal{R}_n(\mathcal{F}_c) \lesssim n^{-\frac{1}{2}}$ and by Equation (18) it follows that: $\mathcal{R}_m(\mathcal{F}_c^c) \lesssim m^{-\frac{1}{2}}$.

Turning now to the Rademacher Complexity of the composition of the c-concave potentials $\mathcal{F}_c$ with $r \circ \mathcal{H}$ (the composition of a fixed reward function with the hypothesis class $\mathcal{H}$ ). Note since the cost $c(., z')$ is $L$ Lipchitiz for all $z' \in [-M, M]$ we have $\mathcal{F}_c$ is included in the set of $L$ lipchitz function that are bounded by $R$ (See [Hundrieser et al., 2022] Lemma A.2 ). For $\varphi \in \mathcal{F}_c$ and $\pi_\theta \in \mathcal{H}$ , let us note $h_{\varphi, \pi_\theta} = \varphi(r \circ \pi_\theta)$ we have:

$$\left\| h_{\varphi, \pi_\theta} - h_{\varphi', \pi_{\theta'}} \right\|_\infty = \sup_{x \in \mathcal{X}, y \in \mathcal{Y}} |\varphi(r \circ \pi_\theta(y|x)) - \varphi'(r \circ \pi_{\theta'}(y|x))|$$

We have:

$$|\varphi(r \circ \pi_\theta(x)) - \varphi'(r \circ \pi_{\theta'})| = |\varphi(r \circ \pi_\theta(x)) - \varphi(r \circ \pi_{\theta'}) + \varphi(r \circ \pi_{\theta'}) - \varphi'(r \circ \pi_{\theta'})|$$
$$\leq L \|r \circ \pi_\theta - r \circ \pi_{\theta'}\|_\infty + \|\varphi - \varphi'\|_\infty .$$

where we used lipchitzity of $\varphi \in \mathcal{F}_c$ and hence we have:

$$\left\| h_{\varphi, \pi_\theta} - h_{\varphi', \pi_{\theta'}} \right\|_\infty \leq L \|r \circ \pi_\theta - r \circ \pi_{\theta'}\|_\infty + \|\varphi - \varphi'\|_\infty .$$

We have therefore the following bound on the covering number of the composition:

$$\mathcal{N}(\varepsilon, \mathcal{F}_c \circ r \circ \mathcal{H}, \|.\|_\infty) \leq \mathcal{N}\left(\frac{\varepsilon}{2}, \mathcal{F}_c, \|.\|_\infty\right) \mathcal{N}\left(\frac{\varepsilon}{2L}, r \circ \mathcal{H}, \|.\|_\infty\right),$$

Plugging this in Equation (19) we obtain:

$$\mathcal{R}_n(\mathcal{F}_c \circ r \circ \mathcal{H}) \leq \inf_{\delta \in [0, R]} \left( 2\delta + \sqrt{32} \frac{1}{\sqrt{n}} \int_{\delta/4}^R \sqrt{\log \mathcal{N}(\varepsilon/2, \mathcal{F}_c, \|.\|_\infty) + \log \mathcal{N}\left(\frac{\varepsilon}{2L}, r \circ \mathcal{H}, \|.\|_\infty\right)} d\varepsilon \right)$$

Note that for $a, b > 0$ we have $\sqrt{a + b} \leq \sqrt{a} + \sqrt{b}$, hence we have:

$$\mathcal{R}_n(\mathcal{F}_c \circ r \circ \mathcal{H}) \leq \inf_{\delta \in [0, R]} \left( 2\delta + \sqrt{32} \frac{1}{\sqrt{n}} \int_{\delta/4}^R \sqrt{\log \mathcal{N}(\varepsilon/2, \mathcal{F}_c, \|.\|_\infty)} + \sqrt{\log \mathcal{N}\left(\frac{\varepsilon}{2L}, r \circ \mathcal{H}, \|.\|_\infty\right)} d\varepsilon \right)$$

We know by lipchitizty of the cost and being in one dimension that :

$$\log \mathcal{N}(\varepsilon, \mathcal{F}_c, \|.\|_\infty) \lesssim \varepsilon^{-1}$$

By lipchitizity of $r \circ \pi_\theta$ and using Assumption 3 we have therefore:

$$\log \mathcal{N}\left(\frac{\varepsilon}{2L}, r \circ \mathcal{H}, \|.\|_\infty\right) \leq \log \mathcal{N}\left(\frac{\varepsilon}{2LL'}, B_2(r_0, d_\theta), \|.\|_\infty\right) \leq d_\theta \log \frac{2r_0 L'L}{\varepsilon}$$

We have therefore:

$$\inf_{\delta \in [0, R]} 2\delta + 4\frac{\sqrt{2}}{\sqrt{n}} \int_{\delta/4}^R K_1 \left(\frac{\varepsilon}{2}\right)^{-1/2} d\varepsilon + 4\frac{\sqrt{2}}{\sqrt{n}} \int_{\delta/4}^R \sqrt{d_\theta \log \frac{2r_0 LL'}{\varepsilon}} d\varepsilon \lesssim n^{-\frac{1}{2}}.$$

(For $\delta = 0$, the upper bound is obtained.)

For 2) By assumption 4, there exists $\pi_{\theta^*}$, such that we have for $h = (-x)_+^2$ : $\mathsf{OT}_h \left((r \circ \pi_{\theta^*})_\sharp \mu_+, (r \circ \pi_{\theta^*})_\sharp \mu_-\right) = 0$. $\qquad \square$

# E   AOT paired

Similarly following the relaxation approach described in Section 3 and using the dual representation of the OT problem we can relax the paired stochastic dominance constraint problem given in (FSD paired) to:

$$\min_{\pi_\theta \in \mathcal{H}} \sup_{\varphi \in \mathcal{F}_c} \int \varphi(r \circ \pi_\theta) d\mu - \int \varphi^c(r \circ \pi_{\text{ref}}) d\mu, \tag{20}$$

where $\mu$ is the paired measure representing $(X, Y_+, Y_-)$. let $\pi_{\theta^*}$ be the minimizer of (20). Considering Problem (20), with empirical samples $\hat{\mu}^n = \frac{1}{n} \sum_{i=1}^n \delta_{(x_i, y_{i,+}, y_{i,-})}$ , denote $\pi_{\hat{\theta}_n}$ its minimizer we have :

**Theorem 3** (Sample Complexity of Dominance Violation for AOT Paired). *The following sample complexity bound for the violation of stochastic dominance in* AOT *paired holds:*

$$\mathbb{E} \, \mathsf{OT}_h \left( (r \circ \pi_{\hat{\theta}_n})_\sharp \mu, (r \circ \pi_{\mathrm{ref}})_\sharp \mu \right) \leq \underbrace{\mathsf{OT}_h \left( (r \circ \pi_{\theta^*})_\sharp \mu, (r \circ \pi_{\mathrm{ref}})_\sharp \mu \right)}_{\textit{Optimal Almost FSD Violation}}$$

$$+ \underbrace{2\mathcal{R}_n(\mathcal{F}_c; (r \circ \pi_{\theta^*})_\sharp \mu_+) + 4\mathcal{R}_n(\mathcal{F}_c^c; (r \circ \pi_{\mathrm{ref}})_\sharp \mu)}_{\textit{One dimensional OT sample complexity with optimal } \theta^*}$$

$$+ \underbrace{2\mathcal{R}_n(\mathcal{F}_c \circ r \circ \mathcal{H}; \mu))}_{\textit{Complexity of learning in } \mathcal{H} \textit{ via the 1D OT problem}} ,$$

*where $\mathcal{R}_n(\mathcal{F}; \nu) = \mathbb{E} \sup_{\varphi \in \mathcal{F}} \left| \frac{1}{n} \sum_{i=1}^n \sigma_i \varphi(Z_i) \right|$ is the Rademacher Complexity and for $i = 1 \ldots n$, $\sigma_i$ are independent rademacher random variables and $Z_i \sim \nu$ iid.*

Similarly under Assumptions 1, 2 and 3 we have:

$$\mathbb{E} \, \mathsf{OT}_h \left( (r \circ \pi_{\hat{\theta}_n})_\sharp \mu, (r \circ \pi_{\mathrm{ref}})_\sharp \mu \right) - \mathsf{OT}_h \left( (r \circ \pi_{\theta^*})_\sharp \mu, (r \circ \pi_{\mathrm{ref}})_\sharp \mu \right) \lesssim n^{-\frac{1}{2}}.$$

*Proof of Theorem 3.* The proof can be simply obtained by inspecting the proof of Theorem 1, and we omit it. $\square$

**Remark 2.** *Although $\mathsf{OT}_h$ is one dimensional, an entropic regularization of $\mathsf{OT}_h$ has computational advantages as discussed in Section 3. Results from [Groppe and Hundrieser, 2023] can be leveraged to obtain sample complexity bounds and we obtain under our assumptions also a parametric rate of $n^{-\frac{1}{2}}$. The main insight in [Groppe and Hundrieser, 2023] is in introducing for an entropic regularization parameter $\varepsilon > 0$, the smoothed $(c, \varepsilon, \mu)$ transform and replacing the spaces $\mathcal{F}_c$ by $\mathcal{F}_{c,\varepsilon}$. In the 1D case, up to constants, these spaces have the same covering numbers.*

# F Choice of violation penalty function $h$

Recall we proposed the following three classes of loss functions $h$ in the main text:

1. **Area of violation ("classification")** Setting $h(x)$ to be the 0-1 loss ($\mathbb{1}_{x<0}$) measures the fraction of the interval $[0, 1]$ where a violation occurs, paralleling classification losses which count the number of misclassification.

2. **Wasserstein-1 violation** Setting $h(x)$ to be the hinge loss $(-x)_+$ reduces to measuring the Wasserstein-1 distance from $U_\theta$ to the nearest distribution that has FSD over $V_\theta$.

3. **Wasserstein-2 violation** Setting $h(x)$ to be the squared hinge loss $(-x)_+^2$ reduces to measuring the Wasserstein-2 distance from $U_\theta$ to the nearest distribution that has FSD over $V_\theta$.

Besides measuring different quantities, the optimization-theoretic properties of each are different:

1. **The 0-1 loss** This loss does not penalize the size of the violations, making gradient-based optimization difficult as large violations have no gradient. Additionally, if FSD were not achievable (e.g. a strong teacher policy), not penalizing the size of the violations could result in risky policies.

2. **The hinge loss**, i.e. the Wasserstein-1 violation measure. By its nature, the gradient of small violations is just as large as the gradient of large violations, which may be beneficial for convergence to an FSD result. When FSD is impossible to achieve, this loss will also have the effect of encouraging sparse violations while still penalizing the size of the violations, similar to the L1 norm in the classic Lasso algorithm. **Smooth relaxations** Smooth relaxations of the hinge, e.g., the logistic loss described in the main text, have a nonzero gradient at zero and continue to have a gradient for small positive values. This has the benefit of encouraging quantiles to continue improving after surpassing those of the reference.

3. **The squared hinge loss**, i.e. the Wasserstein-2 violation measure. As a quadratic loss, it no longer prefers sparse violations (c.f. L2 norm regularization). Indeed, the gradient signal vanishes as the violation becomes small, meaning that dense violations are likely and slow to be removed. A potential failure mode would be that the new policy has a small violation, but none of the quantiles outperform the baseline. **Relaxation** Introducing a bias $\beta$ as in the main text addresses this issue, ensuring the gradient at 0 is nonzero.

## G Gradients of the sorting-based objective (11)

We here repeat the sorting-based objective (11) that we propose, adding $n$ as a superscript for clarity.

$$\min_{\theta \in \Theta} \mathsf{OT}_h(\hat{\mu}_{U_\theta}^{(n)}, \hat{\mu}_{V_\theta}^{(n)}) = \min_{\theta \in \Theta} \frac{1}{n} \sum_{i=1}^{n} h(u_\theta^{(i)} - v_\theta^{(i)}). \tag{21}$$

We use gradient-based optimization approaches in practice to find a minimizing $\theta$. We have the following theorem showing the gradients are asymptotically unbiased (and thus friendly to stochastic gradient methods).

**Theorem 4.** *Let $h' = dh/dt$ be L-lipschitz and the gradients of $u_\theta$ and $v_\theta$ with respect to $\theta$ have 1-norm bounded by $M$, i.e. $\|\nabla_\theta u_\theta\|_1 \leq M, \|\nabla_\theta v_\theta\|_1 \leq M$ for all $\theta \in \Theta$. Suppose further that the support of $U_\theta$, $V_\theta$ are bounded and their distributions have densities (i.e. are atomless).[3] Then the gradient of the objective in (21) is asymptotically unbiased as $n \to \infty$.*

*Proof.* Observe that the gradients take the form

$$\nabla_\theta \mathsf{OT}_h(\hat{\mu}_{U_\theta}^{(n)}, \hat{\mu}_{V_\theta}^{(n)}) = \nabla_\theta \frac{1}{n} \sum_{i=1}^{n} h(u_\theta^{(i)} - v_\theta^{(i)}) \tag{22}$$

$$= \frac{1}{n} \sum_{i=1}^{n} h'(u_\theta^{(i)} - v_\theta^{(i)}) \nabla_\theta (u_\theta^{(i)} - v_\theta^{(i)}). \tag{23}$$

Note that as described in the main text, here, the current gradient is determined by the current state of the sorting of the samples, but it is not necessary to differentiate through the sorting algorithm itself as it changes only discretely.

We wish to understand the convergence of the bias of these gradients as $n \to \infty$.

We can write (noting that the $n$ atoms of the empirical $F_{n, V_\theta}$ are distinct with probability 1)

$$\frac{1}{n} \sum_{i=1}^{n} h'(u_\theta^{(i)} - v_\theta^{(i)}) \nabla_\theta (u_\theta^{(i)} - v_\theta^{(i)}) = \underbrace{\frac{1}{n} \sum_{i=1}^{n} h'(u_\theta^{(i)} - F_{n, V_\theta}^{-1}(F_{n, U_\theta}(u_\theta^{(i)}))) \nabla_\theta (u_\theta^{(i)})}_{I}$$

$$\underbrace{- \frac{1}{n} \sum_{i=1}^{n} h'(F_{n, U_\theta}^{-1}(F_{n, V_\theta}(v_\theta^{(i)})) - v_\theta^{(i)}) \nabla_\theta (v_\theta^{(i)})}_{II}.$$

Let's consider the first term, the analysis for the second term is the same. Note that since $h'$ is $L$-Lipschitz

$$\left| h'\left(u - F_{n, V_\theta}^{-1}(F_{n, U_\theta}(u))\right) - h'\left(u - F_{V_\theta}^{-1}(F_{U_\theta}(u))\right)\right| \leq L \left| F_{n, V_\theta}^{-1}(F_{n, U_\theta}(u)) - F_{V_\theta}^{-1}(F_{U_\theta}(u))\right|. \tag{24}$$

Let

$$I' = \frac{1}{n} \sum_{i=1}^{n} h'(u_\theta^{(i)} - F_{V_\theta}^{-1}(F_{U_\theta}(u_\theta^{(i)}))) \nabla_\theta (u_\theta^{(i)}).$$

---

[3]In our alignment setting, this is should not be an issue as our scores are real-valued.

Note $I'$ is a simple empirical average and hence unbiased in the sense that $\mathbb{E}[I']$ is constant with $n$. Now by (24) we can write

$$\|I - I'\|_1 \leq \frac{L}{n} \sum_{i=1}^{n} \left| F_{n,V_\theta}^{-1}(F_{n,U_\theta}(u_\theta^{(i)})) - F_{V_\theta}^{-1}(F_{U_\theta}(u_\theta^{(i)})) \right| \|\nabla_\theta(u_\theta^{(i)})\|_1 \qquad (25)$$

We seek to bound this quantity as $n \to \infty$. By Cauchy-Schwarz,

$$\|I - I'\|_1^2 \leq L^2 \left( \frac{1}{n} \sum_{i=1}^{n} \left( F_{n,V_\theta}^{-1}(F_{n,U_\theta}(u_\theta^{(i)})) - F_{V_\theta}^{-1}(F_{U_\theta}(u_\theta^{(i)})) \right)^2 \right) \left( \frac{1}{n} \sum_{i=1}^{n} \|\nabla_\theta(u_\theta^{(i)})\|_1^2 \right)$$

$$\leq \frac{L^2 M^2}{n} \sum_{i=1}^{n} \left( F_{n,V_\theta}^{-1}(F_{n,U_\theta}(u_\theta^{(i)})) - F_{V_\theta}^{-1}(F_{U_\theta}(u_\theta^{(i)})) \right)^2$$

$$= \frac{L^2 M^2}{n} \sum_{i=1}^{n} \left( v_\theta^{(i)} - F_{V_\theta}^{-1}(F_{U_\theta}(u_\theta^{(i)})) \right)^2$$

where we've assumed that $\left( \frac{1}{n} \sum_{i=1}^{n} \|\nabla_\theta(u_\theta^{(i)})\|_1^2 \right) \leq M^2$. Observe that $F_{V_\theta}^{-1}(F_{U_\theta}(\cdot))$ is the optimal transport plan from $U_\theta$ to $V_\theta$ and is monotonic nondecreasing, hence $F_{V_\theta}^{-1}(F_{U_\theta}(u_\theta^{(i)}))$ are simply a new set of independently drawn order statistics of $v_\theta$, i.e. $v_\theta^{(i,2)}$.

We thus have

$$\|I - I'\|_1^2 \sim \frac{L^2 M^2}{n} \sum_{i=1}^{n} \left( v_\theta^{(i)} - v_\theta^{(i,2)} \right)^2 \qquad (26)$$

$$= L^2 M^2 \int \left( [F_{n,V_\theta}^{(1)}]^{-1}(t) - [F_{n,V_\theta}^{(2)}]^{-1}(t) \right)^2 dt \qquad (27)$$

where $F_{n,V_\theta}^{(i)}$ are independently realized empirical quantile functions.

Theorem 3.1 of Del Barrio et al. [2018] with the subsequent Remark 3.2.1 therein applies directly to this regime, with the following restated result:

**Theorem 5** (Special case of Theorem 3.1 in light of Remark 3.2.1 in Del Barrio et al. [2018])**.** *If $F$ and $G$ are CDFs of 1-dimensional distributions with bounded support and $G^{-1}$ is continuous on $(0,1)$, then*

$$\int_0^1 \left( F_n^{-1} - G_m^{-1} \right)^2 - \int_0^1 \left( F^{-1} - G^{-1} \right)^2 \to_p 0$$

*as $n, m \to \infty$.*

Applying Theorem 5 to our setting and noting that for us the second integral is zero, we have that if $V_\theta$ has bounded support,

$$L^2 M^2 \int \left( [F_{n,V_\theta}^{(1)}]^{-1}(t) - [F_{n,V_\theta}^{(2)}]^{-1}(t) \right)^2 dt \to_p 0$$

where $\to_p$ indicates convergence in probability as $n \to \infty$. This implies that $I - I'$ converges to 0 in probability. The proof for $II$ is identical. Since the gradient (23) is then a sum of two unbiased terms and two terms that converge in probability to zero, the gradient is asymptotically unbiased. $\qquad \square$

## H  Additional Experiments

Tables 2, 3, 4, 5 and 6 are ablations on the reference base model used (`Merlinite-7B`, `OpenHermes-2.5-Mistral-7B`, `Sarling-LM-7B-alpha`, `Meta-LLama-3-8B-Instruct`, `Mistral-7B-Instruct-v0.2`). In these tables, we report only the AlpacaEval using `Llama3-70B` as a judge to reduce the costs of evaluations. Note that we observed that while the absolute scoring of `Llama3-70B` is different than GPT4, as it can be seen in Table 2, it preserves the rankings of the models. We see across all these models a better performance of the distributional alignment AOT on AlpacaEval and a competitive performance on other benchmarks.

|  | AlpacaEval (Llama3-70B) | AlpacaEval (GPT4) | ARC | Hellaswag | MMLU | Truthful | Winogrande | GSM8K |
|---|---|---|---|---|---|---|---|---|
| AOT paired | 44.3 | 29.9 | 82.5 | 66.1 | 62.9 | 50.8 | 74.4 | 53.1 |
| AOT unpaired | **48.4** | **31.3** | 82.5 | **66.2** | 62.8 | **51.1** | 74.4 | 51.8 |
| DPO | 36.8 | 27.4 | **82.8** | 65.8 | **63.1** | 50.6 | 74.3 | 52.0 |
| KTO | 35.7 | 24.9 | 82.7 | 65.4 | 63.0 | 48.7 | **74.9** | **53.9** |
| IPO | 43.1 | 27.7 | 82.4 | 65.1 | 63.0 | 46.5 | 74.0 | 52.3 |
| Merlinite-7B | 28.8 | 17.1 | 81.6 | 63.2 | 62.6 | 42.0 | 73.9 | 45.2 |

Table 2: Merlinite-7B trained on UltraFeedback Binarized. Here we present full version of the results, including AlpacaEval using Llama3-70B-instruct as a judge and GPT4 as a judge. The comparison reveals that although Llama3 inflates the scores, the relative order between the two judges remains the same, suggesting the use of a cheaper AlpacaEval alternative for local development.

|  | AlpacaEval (Llama3-70B) | ARC | Hellaswag | MMLU | Truthful | Winogrande | GSM8K |
|---|---|---|---|---|---|---|---|
| AOT paired | **24.4** | 84.1 | **66.1** | 61.0 | **50.6** | **74.9** | 66.6 |
| AOT unpaired | 22.5 | **84.2** | 66.0 | 61.0 | 50.5 | 74.8 | 65.7 |
| DPO | 17.9 | 84.1 | 66.0 | 61.0 | 50.4 | 74.4 | **66.7** |
| KTO | 12.6 | 83.5 | 64.3 | **61.1** | 47.2 | 74.4 | 66.3 |
| IPO | 15.5 | 83.9 | 65.4 | **61.1** | 49.2 | 74.2 | 66.3 |
| OpenHermes-7B | 5.6 | 83.4 | 63.1 | 60.6 | 44.5 | 74.4 | 63.8 |

Table 3: OpenHermes-2.5-Mistral-7B trained on UltraFeedback Binarized

|  | AlpacaEval (Llama3-70B) | ARC | Hellaswag | MMLU | Truthful | Winogrande | GSM8K |
|---|---|---|---|---|---|---|---|
| AOT paired | 30.4 | 84.3 | 66.9 | 61.4 | 45.5 | 72.6 | 69.0 |
| AOT unpaired | **34.4** | **85.1** | **67.4** | **61.5** | **47.0** | 72.3 | 68.5 |
| DPO | 28.6 | 84.5 | 66.7 | 61.4 | 45.3 | 72.5 | 69.8 |
| KTO | 27.2 | 84.8 | 67.0 | 61.4 | 46.2 | **74.2** | **70.2** |
| IPO | 28.6 | 84.5 | 66.7 | 61.4 | 44.4 | 72.9 | 69.8 |
| Starling-7B | 14.3 | 83.4 | 64.4 | 60.9 | 39.4 | 72.5 | 66.6 |

Table 4: Starling-LM-7B-alpha trained on UltraFeedback Binarized

| | AlpacaEval (Llama3-70B) | ARC | Hellaswag | MMLU | Truthful | Winogrande | GSM8K |
|---|---|---|---|---|---|---|---|
| AOT paired | 33.6 | 81.7 | 59.4 | **64.1** | 47.7 | 72.6 | **78.0** |
| AOT unpaired | **35.8** | 81.8 | 59.4 | 64.0 | **47.8** | 72.8 | 77.5 |
| DPO | 33.1 | **82.1** | **59.5** | 64.0 | 47.3 | 73.1 | 77.9 |
| KTO | 28.5 | 81.9 | 59.0 | 63.9 | 46.5 | **73.4** | 77.7 |
| IPO | 33.2 | 81.9 | 59.1 | 63.9 | 46.7 | 72.9 | 77.7 |
| Llama3-8B | 25.4 | 81.6 | 57.7 | 63.8 | 43.9 | 72.5 | 75.9 |

Table 5: Meta-Llama-3-8B-Instruct trained on UltraFeedback Binarized

| | AlpacaEval (Llama3-70B) | ARC | Hellaswag | MMLU | Truthful | Winogrande | GSM8K |
|---|---|---|---|---|---|---|---|
| AOT paired | 32.8 | 81.6 | 67.6 | **58.9** | 62.7 | 74.3 | 41.9 |
| AOT unpaired | **34.3** | 81.7 | 67.6 | 59.1 | **63.0** | **74.4** | 41.9 |
| DPO | 28.8 | 81.7 | 67.6 | 58.8 | 62.3 | 74.2 | **42.0** |
| KTO | 27.4 | **81.9** | **67.7** | 58.8 | 62.5 | 74.3 | 41.7 |
| IPO | 28.4 | **81.9** | 67.1 | 58.8 | 60.5 | 74.0 | 41.9 |
| Mistral-7B | 25.6 | 81.3 | 66.0 | 58.8 | 59.7 | 74.1 | 41.7 |

Table 6: Mistral-7B-Instruct-v0.2 trained on UltraFeedback Binarized

Table 7 is an ablation on the sorting that is used in AOT with `Merlinite-7B` as a reference model. We see that hard and soft sorting are on par in terms of overall performance.

| | Sort Type | AlpacaEval (Llama3-70B) | ARC | Hellaswag | MMLU | Truthful | Winogrande | GSM8K |
|---|---|---|---|---|---|---|---|---|
| AOT paired | Soft | 44.3 | 82.5 | 66.1 | 62.9 | 50.8 | 74.4 | 53.1 |
| | Hard | 43.8 | 82.7 | 66.2 | 62.9 | 50.7 | 74.5 | 53.9 |
| AOT unpaired | Soft | 48.4 | 82.5 | 66.2 | 62.8 | 51.1 | 74.4 | 51.8 |
| | Hard | 49.2 | 82.5 | 65.9 | 62.8 | 51.0 | 74.4 | 51.0 |

Table 7: The effect of sort type on performance in AOT alignment

Tables 8 and 9 give a comparison between AOT and KTO on unpaired datasets (HelpSteer and PKU binarized) we see that overall AOT leads to a better performance than KTO.

| | ARC | Hellaswag | MMLU | Truthful | Winogrande | GSM8K |
|---|---|---|---|---|---|---|
| AOT unpaired | **82.0** | **63.5** | **62.9** | **45.6** | **74.9** | 48.0 |
| KTO | 81.9 | 63.5 | 62.7 | 45.0 | 74.2 | **48.5** |
| Merlinite-7B | 81.6 | 63.2 | 62.6 | 42.0 | 73.9 | 45.2 |

Table 8: Merlinite-7B trained on unpaired HelpSteer (binarized)

| | ARC | Hellaswag | MMLU | Truthful | Winogrande | GSM8K |
|---|---|---|---|---|---|---|
| AOT unpaired | 82.0 | **64.1** | **63.0** | **56.3** | **74.6** | 49.7 |
| KTO | **82.1** | 63.5 | 62.9 | 43.5 | 74.5 | **50.4** |
| Merlinite-7B | 81.6 | 63.2 | 62.6 | 42.0 | 73.9 | 45.2 |

Table 9: Merlinite-7B trained on unpaired PKU (binarized)

Finally Figure 4 puts in context our best model `Merlinite-7B-uAOT` as the best 7B-family model on AlpacaEval leaderboard at the time of writing this paper. Finally, we give in Table 10 the variance of the evaluation across 4 different random seeds for training and evaluation each alignment strategy, we see very small variance in AOT, especially the unpaired variant.

| Version: | AlpacaEval | AlpacaEval 2.0 | Filter: | Community | Verified |

Baseline: GPT-4 Preview (11/06)  |  Auto-annotator: GPT-4 Preview (11/06)

| Model Name | LC Win Rate | Win Rate |
|---|---|---|
| Aligner 2B+GPT-4 Turbo (04/09) | 58.3% | 46.8% |
| GPT-4 Omni (05/13) | 57.5% | 51.3% |
| GPT-4 Turbo (04/09) | 55.0% | 46.1% |
| Yi-Large Preview | 51.9% | 57.5% |
| Storm-7B (num_beams=10) | 51.8% | 55.4% |
| GPT-4 Preview (11/06) | 50.0% | 50.0% |
| Storm-7B | 48.9% | 52.5% |
| ExPO + Llama-3-Instruct-8B-SimPO | 45.8% | 40.6% |
| Llama-3-Instruct-8B-SimPO | 44.7% | 40.5% |
| Nanbeige Plus Chat v0.1 | 44.5% | 56.7% |
| Qwen1.5 110B Chat | 43.9% | 33.8% |
| Aligner 2B+Claude 3 Opus | 41.8% | 34.5% |
| Claude 3 Opus (02/29) | 40.5% | 29.1% |
| GPT-4 | 38.1% | 23.6% |
| Aligner 2B+Qwen1.5 72B Chat | 36.7% | 31.8% |
| Qwen1.5 72B Chat | 36.6% | 26.5% |
| GPT-4 (03/14) | 35.3% | 22.1% |
| Ein 70B v0.1 | 35.0% | 24.8% |
| Claude 3 Sonnet (02/29) | 34.9% | 25.6% |
| FsfairX-Zephyr-Chat-v0.1 | 34.8% | 35.9% |
| Llama 3 70B Instruct | 34.4% | 33.2% |
| Mistral Large (24/02) | 32.7% | 21.4% |
| ExPO + SPPO-Mistral7B-PairRM | 31.8% | 35.4% |
| merlinite-7B-AOT | 31.7% | 29.9% |
| Samba CoE v0.2 (best-of-16) | 31.5% | 27.0% |
| REBEL-Llama-3-8B-Instruct | 31.4% | 34.3% |
| Mixtral 8x22B v0.1 | 30.9% | 22.2% |
| SPPO-Mistral7B-PairRM | 30.5% | 32.2% |
| GPT-4 (06/13) | 30.2% | 15.8% |
| Snorkel (Mistral-PairRM-DPO+best-of-16) | 30.0% | 34.9% |

Figure 4: Our AOT algorithm gives a strong boost to Merlinite-7B model on AlpacaEval leaderboard (as of May 22nd, 2024). The original Merlinite-7B score is 17.1, and after the alignment, the model gained 83%.

|              | AlpacaEval (Llama3-70B) |
| ------------ | ----------------------- |
| AOT paired   | $46.8 \pm 1.48$         |
| AOT unpaired | $48.1 \pm 0.35$         |
| DPO          | $39.2 \pm 2.35$         |
| KTO          | $33.8 \pm 1.23$         |
| IPO          | $45.7 \pm 1.56$         |
| Merlinite-7B | $28.8$                  |

Table 10: Merlinite-7B trained on UltraFeedback Binarized. We evaluated the stability (variance) of the model evaluation on AlpacaEval by running 4 separate training and evaluation cycles, then computing the mean and standard deviations. The results are stable, especially for AOT unpaired, showing a low deviation from the mean.

