# OpenReview forum: "Distributional Preference Alignment of LLMs via Optimal Transport"
_NeurIPS.cc/2024/Conference — NeurIPS 2024 poster_

### Official Review · Reviewer_BcRY · 2024-06-25

**Soundness:** 4
**Presentation:** 4
**Contribution:** 4
**Rating:** 8
**Confidence:** 3

**Summary:**

The paper proposes a new technique for preference alignment, named Alignment via Optimal Transport (AOT). The proposed technique supports both paired and unpaired alignment settings.  The paper introduces a new viewpoint for preference alignment based on stochastic dominance i.e., making the reward distribution of the positive samples stochastically dominant in the first order on the distribution of negative samples. From this perspective, the paper explains DPO as a special case i.e., a pointwise preference approach with relaxation through the logistic loss. In addition, the paper shows that using convex relaxation is equivalent to minimizing a one-dimensional optimal transport problem. To enhance the differentiability of the objective through the sorting operator (for approximating the continuous optimal transport), the paper uses the SInkhorn-Knopp algorithm instead of the conventional sorting. On the experimental side, AOT leads to state-of-the-art models in the 7B family of models when evaluated with Open LLM Benchmarks and AlpacaEval (using Llama3-70B-Instruct instead of GPT4).

**Strengths:**

* The paper is well-written and easy to follow.
* The paper proposes an original preference alignment approach based on stochastic dominance and optimal transport. The connection to optimal transport is interesting and novel. In addition, the paper also provides theoretical results on the sample complexity of the empirical estimation of the objective.
* The proposed approach can work for both unpaired and paired alignment settings.
* Experiments are extensive on various datasets i.e., UltraFeedback for paired setting,  PKU BeaverTails, and HelpSteer for unpaired setting. The paper also provides a free LLM-judge for local Alpaca evaluations.
* ATO achieves SOTA on AlpacaEval benchmark and competitive results on other metrics compared to DPO, KPO, and IPO.

**Weaknesses:**

The evaluation is conducted using Llama3-70B-Instruct instead of GPT4 which is not the standard. However, as described in the paper, the usage of  Llama3-70B-Instruct  leads to approximately the same.

**Questions:**

1. It seems that hard sorting is also comparable to soft sorting. Using normal sorting could help to improve the computational speed compared to soft sorting. Should the paper recommend hard sorting as the default variant?

2. How many interactions are used for the Sinkhorn algorithm? What is the choice of the entropic regularization hyperparameter? Do these hyperparameters significantly affect the results?

3. Why does the unpaired setting lead to better results than the paired setting? Do the authors have any explanation?

**Limitations:**

The authors adequately addressed the limitations.

---

> ### Author Rebuttal · Authors · 2024-08-05
>
> We thank the reviewer for their positive review of the paper and for their insightful questions that we address below:
> ___
>
> **The evaluation is conducted using Llama3-70B-Instruct instead of GPT4 which is not the standard. However, as described in the paper, the usage of Llama3-70B-Instruct leads to approximately the same**
>
> Thanks for this comment, we have also reported GPT4 eval on Merlinite, and for the rest we used Llama 70B to lower evaluation costs as mentioned by the reviewer.
> ___
>
> **It seems that hard sorting is also comparable to soft sorting. Using normal sorting could help to improve the computational speed compared to soft sorting. Should the paper recommend hard sorting as the default variant?**
>
> We recommend hard sorting as a default since soft sorting did not lead to substantial improvements. And as noted by the reviewer, hard sorting is also faster.
> ___
>
> **How many interactions are used for the Sinkhorn algorithm? What is the choice of the entropic regularization hyperparameter? Do these hyperparameters significantly affect the results?**
>
> The entropic regularization has to be small ~ 0.1 to ensure accurate sorting otherwise the sorting will be wrong. Note that the soft [sorting paper](https://arxiv.org/pdf/2002.08871) uses PAV (Pool adjacent violators) to compute the soft sorting, which, unlike Sinkhorn, does not need a fixed number of iterations as it reduces finding the soft sorting to an isotonic regression that can be computed in closed form with an algorithm $O(n \log n)$.
> ___
>
> **Why does the unpaired setting lead to better results than the paired setting? Do the authors have any explanation?**
>
> Please note that AOT unpaired outperforms AOT paired on AlpacaEval because it is more robust to noise, if some paired positive and negative answers are noisy, AOT paired will fit to that noise, while AOT unpaired since it does not use this pairing will be more robust to noise.  Hence in Figure 2 we see that AOT unpaired outperforms the paired one on AlpacaEval.

---

> > ### Comment · Reviewer_BcRY · 2024-08-09
> > **Response to authors**
> >
> > Thank you for your response,
> >
> > I will keep my score since I believe this paper proposes an interesting framework for LLM alignment i.e., stochastic dominance and optimal transport.
> >
> > Best,

---

### Official Review · Reviewer_9Mcg · 2024-07-11

**Soundness:** 3
**Presentation:** 3
**Contribution:** 2
**Rating:** 6
**Confidence:** 3

**Summary:**

The motivation of this paper is that current alignment approaches ensure reward dominance at the sample-level but not on the distributional level. With this in mind, the authors set their goal to design an alignment approach which satisfies First Order Stochastic Dominance (FSD) in rewards for positive examples compared to negative examples. With this goal in mind, they make the following contributions:
1. They write this as a constraint satisfaction problem, which can be relaxed into an unconstrained optimization problem using different surrogates of the 0/1 loss.
2. Using Santambrogio (2015)'s result from optimal transport, they develop a computationally efficient algorithm called ATO.
3. They provide an upper bound for the violation in stochastic dominance of this algorithm using a Rademacher and symmetrization argument.
4. They provide experiments that show that ATO is competitive with existing approaches like DPO, KTO.

**Strengths:**

**Originality**
1. The connection to optimal transport is quite interesting.

**Quality**

2. The empirical results are competitive with other alignment approaches.
3. The theoretical results are sound

**Clarity**

4. The paper is well-written with background appropriately introduced and clear explanations.

**Weaknesses:**

1. There could be some more motivation for why the FSD condition is practically desirable over the conditional (on x) dominance condition of DPO from Equation (2). Is this more than a theoretical nicety?

2. There is not a clear improvement in the empirical results over prior approaches, and each approach seems to enjoy success on certain benchmarks -- with the differences between performance quite small in most cases.
This invites the question of in what settings ATO would perform better than other approaches in the literature.

3. The theoretical results do not shed much light on when ATO is likely to perform well. Uniform Convergence results are well-known in recent years to not explain generalization behavior of modern ML models. So, while nice, the insight from Theorem 2 is not super obvious to me.

Minor:
1. Please call the Santambrogio (2015) result something other than a theorem since it is not a contribution of the present work.

**Questions:**

Since KTO also works with unpaired preference data, a more thorough comparison with KTO is warranted:
1. Is there a unique solution that satisfies the FSD condition? If there are multiple solutions, how does ATO break ties? How does KTO break ties?
2. Is the reason that the behavior is different from KTO that they have different inductive biases and hence converge to different policies? Or the variances of the estimators are different, even though they converge to the same?
3. Is there a way to compare Theorem 2 with the statistical properties of KTO/DPO/RLHF? Or any reason to believe that ATO has a statistical advantage?

Despite the weaknesses, I think the connection to optimal transport is interesting and the experiments are sound and comprehensive. I would be happy to adjust my score accordingly if the authors alleviate my concerns through discussion.

**Limitations:**

Yes

---

> ### Author Rebuttal · Authors · 2024-08-05
>
> We thank the reviewer for their thoughtful comments:
> ___
>
> **Weaknesses**
> ___
>
> **1) There could be some more motivation for why the FSD condition is practically desirable over the conditional (on x) dominance condition of DPO from Equation (2). Is this more than a theoretical nicety?**
>
> Thanks for bringing this up we will make the following discussion more salient:
>
> * It is more expensive to collect paired preference datasets, that have for each prompt a preferred and a rejected answer. There are plenty of unpaired datasets in the sense for each prompt we have either a preferred or a rejected answer. AOT and KTO allow the use of these datasets in alignment.
>
> * AOT unpaired by not relying on pairs of rejected / preferred answers but rather on their distributions, is more robust to noise in the pairing. While DPO would suffer on a pointwise comparison from noise in the pairing, AOT unpaired would not.
> ___
> **2) There is not a clear improvement in the empirical results over prior approaches, and each approach seems to enjoy success on certain benchmarks -- with the differences between performance quite small in most cases. This invites the question of in what settings ATO would perform better than other approaches in the literature.**
>
> Please note that the alignment scoring we use is AlpacaEval (instruction following), all other metrics from the Open LLM Leaderboard such as (ARC, MMLU, Winogrande, GSM8K) test that despite the alignment the model retains or improves on the capabilities it has acquired from the pretraining model. To benchmark alignment methods AlpacaEval has been well adopted by the community, and the Open LLM Leaderboard metrics assess whether the alignment deteriorates other capabilities of the LLMs. AOT, while having the highest score on AlpacaEval, is competitive on all other metrics.
> ___
>
> **3) The theoretical results do not shed much light on when ATO is likely to perform well. Uniform Convergence results are well-known in recent years to not explain generalization behavior of modern ML models. So, while nice, the insight from Theorem 2 is not super obvious to me.**
>
> Please note that Theorem 2 shows that despite AOT can be cast as a min/max game between the LLM and the OT potential, the one dimensional nature of the OT problem does not curse the statistical properties of the overall problem. The generalization bound we provide enjoys the parametric rate $1/\sqrt{n}$. Note that in Figure 3, we see that the generalization error (on the AlpacEval) improves as the sample size grows which echoes findings in Theorem 2. The main insight of Theorem 2 is while AOT has an inner optimal transport problem to solve in order to update the LLM parameters, this does not curse statistically the rate, since the OT problem is one dimensional.
>
> ___
>
> **Please call the Santambrogio (2015) result something other than a theorem since it is not a contribution of the present work.**
>
> Thank you for the remark, we will take this into consideration in the final version of the paper.  We will add the following sentence before this theorem: The following theorem is a restatement of Theorem 2.9 in Santambrogio (2015).
> ___
>
> **Questions**
> ___
>
> **1) Since KTO also works with unpaired preference data, a more thorough comparison with KTO is warranted:
> Is there a unique solution that satisfies the FSD condition? If there are multiple solutions, how does ATO break ties? How does KTO break ties?**
>
> One can study the uniqueness of the solution in simplified setups where the log-likelihood is a linear model, since the problem becomes convex in the weights, the AOT problem properly regularized with an $L_2$ regularizer over the weight will have a unique solution. Nevertheless, this is not the case for general nonlinear models such as transformers or LoRA adapters when finetuning the LLM,  then multiple solutions may exist.  Studying the connectivity of the minimas with an alignment loss as it is done in deep learning with classification objectives is an interesting avenue for future research and that is beyond the scope of the paper. We believe if multiple minima exist they will achieve the same soft violation of FSD (that is the loss of AOT), and those solutions will be equivalent and most likely connected.
> ___
>
> **2) Is the reason that the behavior is different from KTO that they have different inductive biases and hence converge to different policies? Or the variances of the estimators are different, even though they converge to the same?**
>
> We believe they have different cost functions and hence they will converge to different policies. While AOT compares quantiles of rewards between positive and negative rewards, KTO compares positive rewards to the average negative reward, and negative rewards to the average positive rewards. KTO will lead to a policy that favors sentences whose reward is above the average of negative rewards in the training set, and AOT will lead to a policy that distributionally prefers positive over negative responses.
> ___
>
> **3) Is there a way to compare Theorem 2 with the statistical properties of KTO/DPO/RLHF? Or any reason to believe that ATO has a statistical advantage?**
>
> Statistically speaking KTO, DPO, and AOT have similar statistical rates. AOT does not suffer statistically from having to solve an OT problem since this is a one dimensional problem, and hence still has a good statistical rate while being a min/max problem between the LLM and the potential of the OT problem (the sorting). For RLHF the statistical properties are more intricate since there is the reward learning and the sampling from the policy in training. The following paper gives a statistical analysis of a [variant of RLHF](https://arxiv.org/pdf/2405.21046 ). Different from our analysis this RLHF variant has also regret bounds that scale like $1/\sqrt{T}$.

---

> > ### Comment · Reviewer_9Mcg · 2024-08-13
> > **Rebuttal response**
> >
> > I thank the authors for clarifying my questions, and accordingly increase my score to 6.

---

### Official Review · Reviewer_rub8 · 2024-07-13

**Soundness:** 3
**Presentation:** 3
**Contribution:** 2
**Rating:** 4
**Confidence:** 2

**Summary:**

This works proposed using Optimal transport in 1D to derive a better alignment guided by the preference data. The main idea is to work with the log likelihood  ratio  of  the  marginal distributions of the preference data. The alignment is made through Optimal Transport loss in 1D based on the concept of stochastic dominance between quantiles of two distributions.

**Strengths:**

The framework of the paper is very easy to follow.
The proposed problem and solution's direction are interesting.
Settings, theoretical results are adequate to support the proposed method.

**Weaknesses:**

My main concern of this paper is the empirical results in experiment section. Table 1 shows that AOT paired/unpaired do not outperform other methods at least in 4 out 7 cases (ARC, MMLU, Winogrande, GSM8K). When they are versus each other, there is no clear winner between AOT paired and AOT unpaired. Meanwhile, I believe that with the AOT paired, when we have more information, the task must be easier, please correct me if I am wrong. I have similar concerns for their performances in Figure 2.

**Questions:**

In Figure 3, it appears that small $\beta$ will produce better results for all method except the IPO. Have the authors tried to test with smaller value of $\beta$, i.e. $\beta = 0.005$?  Is there any explanation for this trend in that figure?

**Limitations:**

It is fine.

---

> ### Author Rebuttal · Authors · 2024-08-05
>
> We thank the reviewer for their encouraging feedback.
> ___
> **My main concern of this paper is the empirical results in experiment section. Table 1 shows that AOT paired/unpaired do not outperform other methods at least in 4 out 7 cases (ARC, MMLU, Winogrande, GSM8K). When they are versus each other, there is no clear winner between AOT paired and AOT unpaired. Meanwhile, I believe that with the AOT paired, when we have more information, the task must be easier, please correct me if I am wrong. I have similar concerns for their performances in Figure 2.**
>
> Please note that the alignment scoring we use is AlpacaEval (instruction following), all other metrics from the Open LLM Leaderboard such as  (ARC, MMLU, Winogrande, GSM8K)  test that despite the alignment the model retains or improves on the capabilities it has acquired from the pretraining model. To benchmark alignment methods AlpacaEval has been well adopted by the community, and the Open LLM Leaderboard metrics assess whether the alignment deteriorates other capabilities of the LLMs. AOT, while having the highest score on AlpacaEval, is competitive on all other metrics.
>
> Please note that AOT unpaired outperforms AOT paired on AlpacaEval because it is more robust to noise, if some paired positive and negative answers are noisy, AOT paired will fit to that noise, while AOT unpaired, since it does not use this pairing, will be more robust to noise.  While in AOT paired the task may seem easier, it is less robust. Hence in Figure 2 we see that AOT unpaired outperforms the paired one on AlpacaEval.
> ___
>
> **$\beta$ will produce better results for all method except the IPO. Have the authors tried to test with smaller value of $\beta$, i.e.  $\beta=0.005$**
>
> Please note this is an artifact of the formulation of IPO and its implementation in [hugging face](  https://github.com/huggingface/trl/blob/main/trl/trainer/dpo_trainer.py#L1145C1-L1146C1) while for all other methods $\beta$ plays the role of a margin 1/$\beta$ plays the role of the margin in IPO. This explains this trend in Figure 3, IPO peaks on high values of beta and others on lower values, note this same trend has been also observed in the [hugging face blog]( https://huggingface.co/blog/pref-tuning) comparing these alignment techniques.

---

> > ### Comment · Reviewer_rub8 · 2024-08-13
> > **Reply to the rebuttal**
> >
> > I would like to thank the authors for their answers. I would like to keep my score unchanged based on the fact is that  the empirical results on other metrics are only competitive, although the theory appears to be sufficient.

---

### Comment · Area_Chair_ZPQx · 2024-08-12
**Discussion with the authors**

Dear reviewers:

As the discussion period is going to end soon, please try to actively engage with the authors about the paper. Thanks a lot for your help and dedication.

You AC.

---

### Decision · Program_Chairs · 2024-09-25

**Decision:**

Accept (poster)

**Comment:**

The paper introduces Alignment via Optimal Transport (AOT), a new method for aligning large language models (LLMs) based on distributional preferences rather than pairwise human preferences. AOT ensures that the reward distribution of positive samples stochastically dominates that of negative samples by solving an optimal transport problem with a smooth, convex cost, leading to a closed-form solution. By fine-tuning LLMs with the AOT objective, the method achieves superior alignment, demonstrated through state-of-the-art performance on various alignment datasets and benchmarks, including Open LLM Benchmarks and AlpacaEval. All reviewer recognized that the paper has some interesting theoretical contributions which connects optimal transport to the alignment problem. However, the numerical evaluation of this work seems to be less adequate, which the authors can improve upon.